# *PACSIN2* as a modulator of autophagy and mercaptopurine cytotoxicity: mechanisms in lymphoid and intestinal cells

Giulia Zudeh[1], Raffaella Franca[2], Marianna Lucafò[1], Erik J Bonten[3], Matteo Bramuzzo[4], Riccardo Sgarra[5], Cristina Lagatolla[5], Martina Franzin[1], William E Evans[6], Giuliana Decorti[1,2], Gabriele Stocco[1,2]

PACSIN2 variants are associated with gastrointestinal effects of thiopurines and thiopurine methyltransferase activity through an uncharacterized mechanism that is postulated to involve autophagy. This study aims to clarify the role of PACSIN2 in autophagy and in thiopurine cytotoxicity in leukemic and intestinal models. Higher autophagy and lower PACSIN2 levels were observed in inflamed compared with non-inflamed colon biopsies of inflammatory bowel disease pediatric patients at diagnosis. PACSIN2 was identified as an inhibitor of autophagy, putatively through inhibition of autophagosome formation by a protein–protein interaction with LC3-II, mediated by a LIR motif. Moreover, PACSIN2 resulted a modulator of mercaptopurine-induced cytotoxicity in intestinal cells, suggesting that PACSIN2-regulated autophagy levels might influence thiopurine sensitivity. However, PACSIN2 modulates cellular thiopurine methyltransferase activity via mechanisms distinct from its modulation of autophagy.

## Introduction

Thiopurines, such as mercaptopurine and azathioprine, are purine analogues used in the treatment of hematological malignancies (i.e., acute lymphoblastic leukemia, ALL) and inflammatory conditions (i.e., inflammatory bowel diseases, IBD) (Bermejo et al, 2018; Lamb et al, 2019; Relling et al, 2019). Through a complex anabolic enzymatic pathway, thiopurines are biotransformed into thionucleotides that exert the cytotoxic action responsible for treatment efficacy; thiopurines also undergo catabolic processes, in which the enzyme thiopurine S-methyltransferase (TPMT) plays a major role by transforming mercaptopurine into the inactive methylmercaptopurine metabolite (Zaza et al, 2010). TPMT activity is an important determinant of severe adverse events during treatment with thiopurines (Relling & Evans, 2015; Lucafò et al, 2018). Genetic polymorphisms in the *TPMT* gene are major determinants of reduced TPMT activity, but there is substantial unexplained variability in TPMT activity in patients who inherit two wild-type *TPMT* alleles (Tamm et al, 2017). We previously showed that the TT genotype in the protein kinase C and casein kinase substrate in neurons 2 (*PACSIN2*) rs2413739 (C > T) polymorphism modulates TPMT activity and mercaptopurine-induced toxicity, with a molecular mechanism that remains unclear (Stocco et al, 2012; Franca et al, 2020). In particular, the presence of the *PACSIN2* TT genotype in ALL patients was associated with reduced TPMT activity during the maintenance phase of treatment and also with the incidence of severe gastrointestinal toxicities during the consolidation therapy. The association between *PACSIN2* rs2413739 T variant and TPMT activity was not confirmed in a cohort of IBD pediatric patients undergoing azathioprine therapy; however, IBD pediatric carriers of the T allele presented reduced azathioprine effectiveness (Stocco et al, 2012; Franca et al, 2020). Furthermore, a positive correlation between PACSIN2 and TPMT protein concentration was detected in peripheral blood mononuclear cells of healthy donors, further supporting a possible role of *PACSIN2* in TPMT regulation (Franca et al, 2020). Other authors identified the *PACSIN2* rs2413739 TT genotype as a significant risk factor for the development of mercaptopurine-induced hematological toxicity in ALL pediatric patients presenting the wild-type *TPMT* genotype (Smid et al, 2016).

PACSIN2, also called syndapin II, is a protein involved in membrane remodeling pathways containing a FBAR domain (Kessels & Qualmann, 2004) that has been shown to inhibit vesicle formation by protein–protein interaction, influencing processes such as endocytosis (Modregger et al, 2000; de Kreuk et al, 2012) and caveolae formation (Hansen et al, 2011; Senju et al, 2015). Recently, it was demonstrated that PACSIN2 plays a role in the intestinal apical microvillus formation (Postema et al, 2019). Previous data demonstrated

[1]Department of Translational and Advanced Diagnostics, Institute for Maternal and Child Health I.R.C.C.S. Burlo Garofolo, Trieste, Italy    [2]Department of Medical, Surgical and Health Sciences, University of Trieste, Trieste, Italy    [3]Department of Chemical Biology and Therapeutics, Saint Jude Children's Research Hospital, Memphis, TN, USA    [4]Department of Gastroenterology, Digestive Endoscopy and Nutrition Unit, Institute for Maternal and Child Health I.R.C.C.S. Burlo Garofolo, Trieste, Italy    [5]Department of Life Sciences, University of Trieste, Trieste, Italy    [6]Department of Pharmaceutical Sciences, Saint Jude Children's Research Hospital, Memphis, TN, USA

Correspondence: decorti@units.it

that the activity of overexpressed TPMT was reduced after *PACSIN2* knockdown (KD) in a human B-lineage lymphoblastic leukemia cell line (NALM6). The agnostic gene expression analysis of NALM6 with *PACSIN2* (KD) identified autophagy as one of the pathways significantly affected by the reduction in *PACSIN2*, without indicating the direction of this possible effect or confirming these results with mechanistical analyses (Stocco et al, 2012). Previous studies showed that PAC-SIN1 is involved in the regulation of autophagy machinery (Szyniarowski et al, 2011; Oe et al, 2022). Autophagy is a conserved lysosome-dependent cellular degradation program that responds to different environmental and cellular stresses, such as the accumulation of unfolded protein aggregates, the presence of dysfunctional organelles, and the presence of intracellular pathogens (Chun & Kim, 2018). Autophagy is an important mechanism to maintain tissue homeostasis, and its tight regulation is fundamental to prevent pathogenic conditions (Mizushima et al, 2008). Indeed, impaired autophagy is associated with the development of many pathological conditions, including IBD (Iida et al, 2017). In intestinal epithelial cells, autophagy acts as a protective mechanism against cell death and inflammation (Tang et al, 2011; Lapaquette et al, 2015) through different molecular mechanisms that, however, are not fully understood yet (Saitoh et al, 2008). Autophagy begins with the formation of an autophagosome, a double-membrane vesicle incorporating molecules destined for degradation; after autophagosome maturation, these vesicles fuse with lysosomes and the content will be degraded (Zhao & Zhang, 2019). Autophagy is usually investigated evaluating the cellular amount of both the microtubule-associated protein light chain 3 (LC3) protein and the sequestosome-1 (SQSTM1/P62) protein that are considered two important autophagic markers (Klionsky et al, 2021). LC3 is a protein involved in autophagosome elongation and maturation, whereas SQSTM1/P62 is involved in the receptor-mediated autophagy and serves as a link between LC3 and ubiquitinated cargoes to generate the autophagosome around molecules destined for degradation (Tanida et al, 2008; Liu et al, 2016). Different autophagy modulators and receptors, such as SQSTM1/P62, exert their function on autophagy through an LC3-interacting region (LIR) motif (Pankiv et al, 2007; Cadwell et al, 2008).

This study aims to clarify the role of PACSIN2 in autophagy and in thiopurine cytotoxicity in leukemic and intestinal models. Moreover, we hypothesized that the modulation of *PACSIN2* on TPMT activity and protein expression could be due to its role as a modulator of autophagy. We demonstrated a role of PACSIN2 as a negative regulator of autophagy, showing an increase in the basal autophagy level after *PACSIN2* KD in cell lines. Consistent results were reported also for primary tissues: lower *PACSIN2* mRNA and protein expression levels corresponded to higher autophagy and inflammation levels in colon samples of a cohort of IBD pediatric patients. PACSIN2 presented one LIR domain, responsible for PACSIN2 interaction with LC3. Furthermore, intestinal cell lines with *PACSIN2* KD were significantly more sensitive to mercaptopurine cytotoxic effects than control cells. Finally, mercaptopurine exposure decreased autophagy and stimulated apoptosis, especially in the presence of *PACSIN2* KD. However, *PACSIN2* KD led to lower TPMT protein concentrations, by mechanisms other than enhanced protein degradation by autophagy.

# Results

## PACSIN2 expression level is increased in autophagy-deficient cells and reduced in autophagy-competent cells upon induction of autophagy

Because the potential role of PACSIN2 in autophagy is largely unknown, we initially compared PACSIN2 protein levels between confluent autophagy-defective MEFs (Atg7$^{-/-}$) and autophagy-proficient wild-type MEFs (Atg7$^{+/+}$). Immunoblotting using a PACSIN2-specific antibody showed that PACSIN2 protein expression was almost two times higher in autophagy-defective cells compared with autophagy-proficient cells (fold change 1.7 ± 0.41, $P$ = 0.049, Fig 1).

We then assessed whether PACSIN2 was degraded by autophagosomes: during treatment with the chemical inducer of autophagy rapamycin, the half-life of PACSIN2 was 34.7 ± 6.98 h in wild-type MEFs, compared with 117.5 ± 25.7 h in autophagy-deficient Atg7$^{-/-}$ cells ($P$ = 0.030, Fig 2), which indicates that in autophagy-competent cells, PACSIN2 is targeted for degradation by the autophagosome.

## Autophagy is increased after *PACSIN2* KD under basal condition

To determine whether PACSIN2 plays a role in autophagy, we performed immunoblotting assays to evaluate protein expression levels of the autophagic markers LC3-II and SQSTM1/P62 in both lymphoid (NALM6) and intestinal (LS180) cell lines (MOCK and *PACSIN2* KD), and also performed confocal microscopy measurement of GFP-LC3 *punctae* in adherent cell lines HeLa $^{GFP-LC3}$ and RAW 264.7 $^{GFP-LC3}$ cells (Figs S1 and S2).

Under basal conditions, LC3-II was significantly more expressed in cells with *PACSIN2* KD compared with control cell lines in both

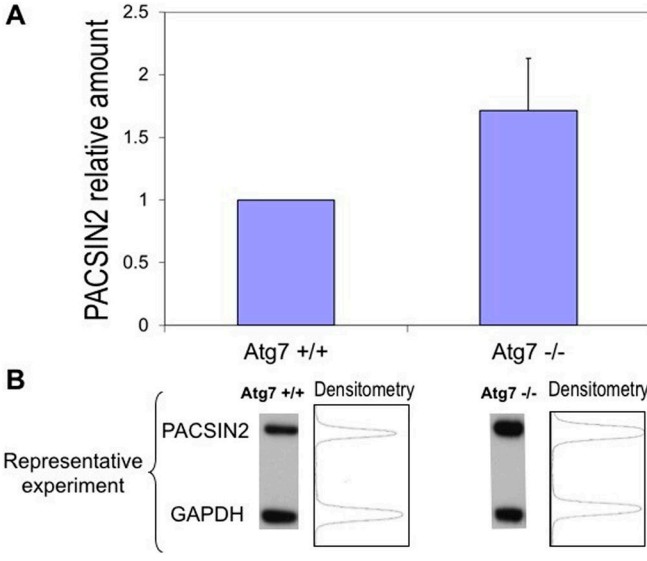

**Figure 1. PACSIN2 protein expression is increased in autophagy-defective cells.**
**(A)** Autophagy-defective cells (Atg7$^{-/-}$ MEFs) express constitutively higher (1.7 ± 0.25 times) endogenous PACSIN2 in comparison with autophagy-proficient cells (Atg7$^{+/+}$) (n = 5, *t* test, *P* = 0.049). **(B)** Representative densitometric analysis on immunoblots of confluent MEF cell lysates.

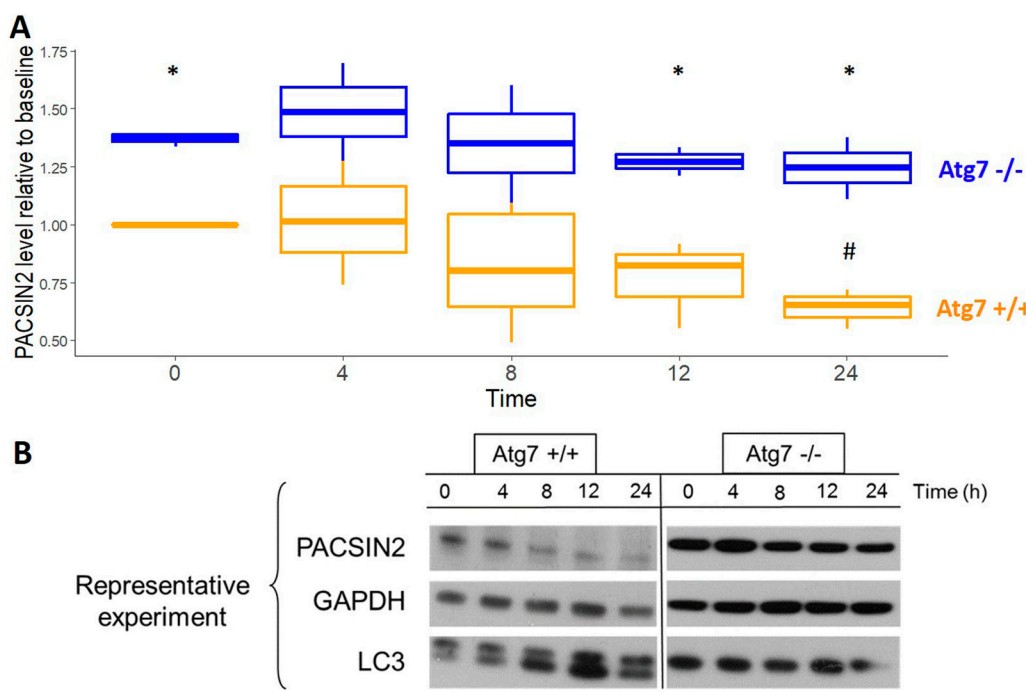

**Figure 2.  PACSIN2 protein expression level is reduced in autophagy-competent cells upon induction of autophagy.**
**(A)** PACSIN2 protein is reduced by induction of autophagy using rapamycin in MEF cells with functional autophagy (Atg7$^{+/+}$, in orange) in comparison with autophagy-defective cells (Atg7$^{-/-}$, in blue); after 24 h, a significantly decreased concentration (n = 3, $t$ test, $P$ = 0.049) of PACSIN2 was measured in Atg7$^{+/+}$, but not in Atg7$^{-/-}$ in comparison with baseline; concentration of PACSIN2 was significantly lower in Atg$^{+/+}$ cells than in Atg7$^{-/-}$ cells at baseline and after 12 (n = 3, $t$ test, $P$ = 0.029) and 24 h (n = 3, $t$ test, $P$ = 0.0045) of treatment, resulting in a half-life for PACSIN2 of 34.7 ± 6.98 h in Atg7$^{+/+}$ cells compared with 117.5 ± 25.7 h in Atg7$^{-/-}$ (n = 3, $t$ test, $P$ = 0.030).
**(B)** Representative image showing protein expression was assessed by immunoblotting on confluent MEFs (n = 3). In these experiments, we used 50 $\mu$g of cell lysates. # = $t$ test, $P$ < 0.05 comparing each cell line PACSIN2 level at a specific time point with baseline (time 0); * = $t$ test, $P$ <0.05 comparing PACSIN2 level between Atg7+/+ and Atg7−/− at each time point.

NALM6 (fold change 2.1 ± 0.18, $P$ = 0.0031, Fig 3A) and LS180 (fold change 1.3 ± 0.09, $P$ = 0.028, Fig 3B). In NALM6, LC3-I was also significantly increased when *PACSIN2* was knocked down (fold change 1.7 ± 0.13, $P$ = 0.0045, Fig 3A). Moreover, both NALM6 KD and LS180 KD cells presented lower SQSTM1/P62 levels compared with MOCK cells (fold change NALM6: 0.67 ± 0.09, $P$ = 0.023, Fig 3C; LS180: 0.47 ± 0.14, $P$ = 0.019, Fig 3D).

Furthermore, we used cells with forced expression of GFP-LC3 and stable *PACSIN2* KD and measured the percentage of autophagic cells as demonstrated by the number of cells containing GFP-LC3 *punctae*. RAW 264.7$^{GFP-LC3}$ KD cells presented an increased basal number of GFP-LC3 *punctae* per cell compared with RAW 264.7$^{GFP-LC3}$ control cells (Fig 4). A similar trend was observed in HeLa$^{GFP-LC3}$ cell line (Fig S3).

The confocal results were in accordance with the increased amount of LC3-II in *PACSIN2* KD cells found by immunoblotting, and taken together, these results provide experimental evidence so far lacking on the involvement of PACSIN2 in the autophagy machinery and describe the direction of this effect, confirming the hypothesis generated by the agnostic approaches used in previous studies.

### *PACSIN2* KD alters the first phases of the autophagic flux

To understand the potential mechanism on the basis of *PACSIN2* regulation of the autophagic flux, LC3-II and SQSTM1/P62

concentrations were evaluated in both NALM6 and LS180 cell lines in the presence of 30 $\mu$M chloroquine, an autophagy inhibitor that impairs the late stage of the autophagic flux, blocking the process when autophagosomes fuse with lysosomes. As expected, 4 h of chloroquine treatment increased LC3-II in both MOCK and KD cell lines, but LC3-II amount was significantly higher in cells with *PACSIN2* KD compared with MOCK in both NALM6 (fold change 1.5 ± 0.13, $P$ = 0.020, Fig 3A) and LS180 (fold change 0.93 ± 0.03, $P$ = 0.035, Fig 3B) cells, suggesting that PACSIN2 could inhibit autophagy acting at the beginning of the autophagic flux. To evaluate the possible impact of chloroquine on SQSTM1/P62, we treated cells with this autophagic inhibitor for 24 h. Chloroquine treatment increased SQSTM1/P62 amount both in NALM6 KD and in LS180 KD cells (fold change NALM6: 1.57 ± 0.1, $P$ = 0.047, Fig 3C; LS180: 3.94 ± 1.8, $P$ = 0.046, Fig 3D) and not in NALM6 and LS180 MOCK cells, supporting the hypothesis that PACSIN2 could play a role in the first phases of autophagy.

### Higher inflammation and autophagy levels correspond to lower *PACSIN2* gene expression and protein concentrations in colon samples of IBD pediatric patients

Because impaired autophagy has been related to the development of pathological conditions, such as IBD, we evaluated the PACSIN2 gene and protein expression levels in inflamed and non-inflamed colon biopsies of IBD pediatric patients at diagnosis. In inflamed

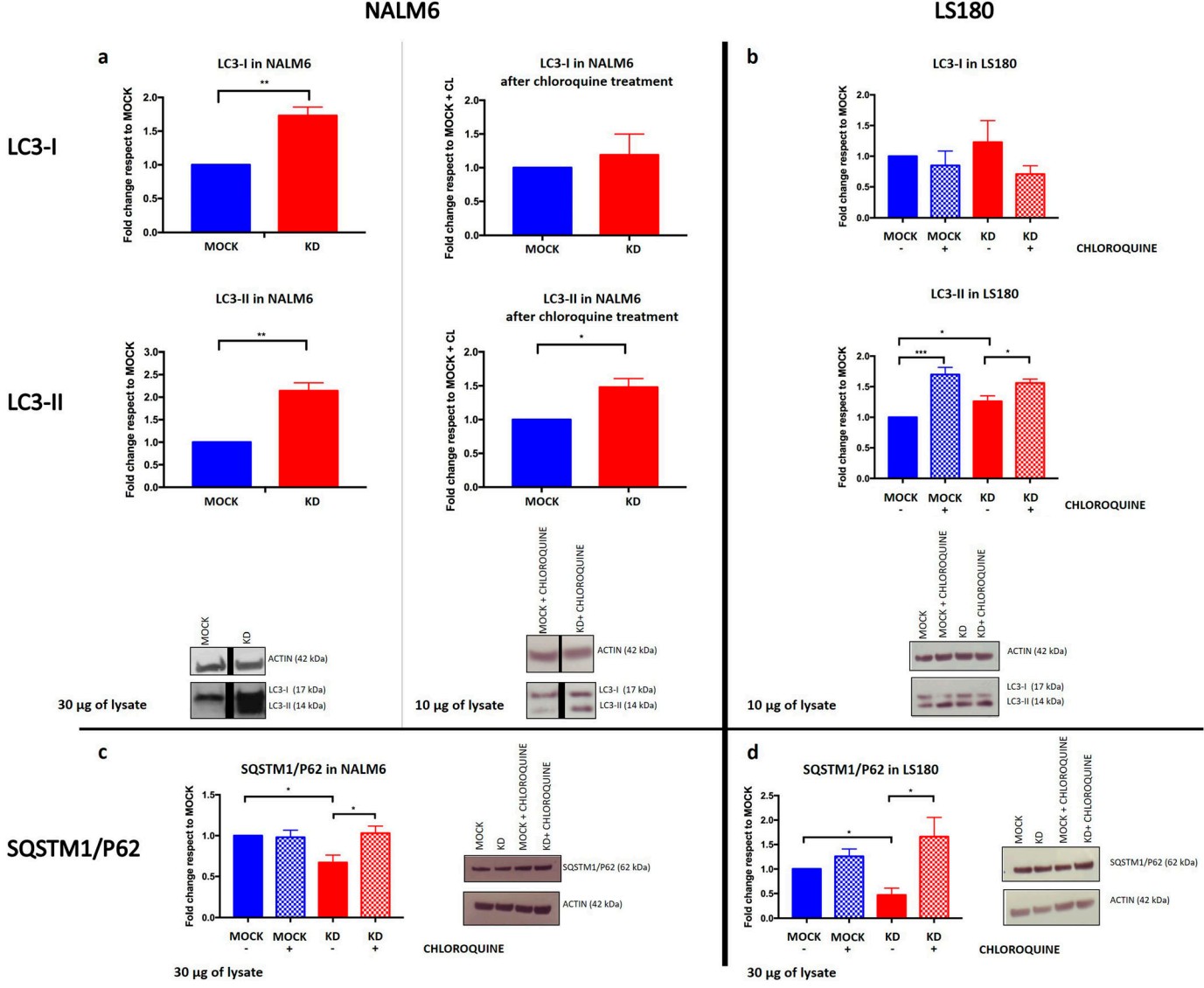

**Figure 3. Autophagy is increased in the presence of *PACSIN2* knockdown (KD) in lymphoid (NALM6) and intestinal (LS180) cell lines.**
**(A)** In NALM6 KD cells, there was an increase in both LC3-I (fold change 1.73 ± 0.127, n = 3, *t* test, *P* = 0.0045) and LC3-II (fold change 2.14 ± 0.179, n = 3, unpaired *t* test, *P* = 0.0031) compared with control cells. After chloroquine treatment, LC3-II levels were increased in both NALM6 MOCK and KD cell lines, and interestingly, LC3 increase was higher in NALM6 KD cells but further in NALM6 KD cells (fold change 1.48 ± 0.127, n = 5, *t* test, *P* = 0.0195). In these immunoblots, 30 µg of cell lysates was used under basal condition, whereas to detect LC3 after chloroquine treatment, the loaded amount of lysates was reduced to 10 µg to avoid signal saturation. The black lines in the blots indicated a splice between lanes. **(B)** LC3-II amount was higher in LS180 KD cells compared with MOCK, under basal condition (fold change 1.26 ± 0.09, n = 4, *t* test, *P* = 0.0277). In the presence of chloroquine treatment, both LS180 MOCK and LS180 KD cells presented higher LC3-II amount and these levels were significantly higher in LS180 KD cells (fold change 0.93 ± 0.03, n = 4, *t* test, *P* = 0.0355). In these experiments, we used 10 µg of cell lysates to evaluate LC3-II amount. **(C)** NALM6 KD cells presented lower SQSTM1/P62 levels compared with control cells under basal condition (fold change 0.62 ± 0.09, n = 3, *t* test, *P* = 0.023); 24 h of chloroquine treatment rescued SQSTM1/P62 levels in NALM6 KD cells (fold change 1.57 ± 0.1, n = 3, *t* test, *P* = 0.047). In these immunoblots, 30 µg of cell lysates was loaded to detect SQSTM1/P62 levels. **(D)** LS180 KD cells presented lower SQSTM1/P62 levels compared with control cells under basal condition (fold change 0.47 ± 0.14, n = 3, *t* test, *P* = 0.018); moreover, 24 h of chloroquine treatment increased SQSTM1/P62 concentrations, particularly in LS180 KD cells (fold change 3.94 ± 1.8, n = 3, *t* test, *P* = 0.046). In these experiments, we used 30 µg of cell lysates to evaluate SQSTM1/P62 amount.
Source data are available for this figure.

tissues, lower amount of both PACSIN2 mRNA (fold change 0.65 ± 0.63, *P* = 0.0084, Fig 5A) and protein (fold change −0.38 ± 0.12, *P* = 0.017, Fig 5B) levels and higher amount of LC3-II protein (fold change 1.32 ± 0.04, *P* = 0.024, Fig 5B) were observed compared with non-inflamed tissues. Furthermore, these results were confirmed by a correlation analysis performed on these intestinal samples

(Pearson's r = −0.83, −0.97% to −0.3 95% confidence interval, *P* = 0.011, Fig S4), showing an inverse correlation between PACSIN2 amount and autophagy levels, supporting the results observed in cell lines, and associating reduced PACSIN2 with increased autophagy. To evaluate the possible impact of the immune system cells, which could be infiltrated in the intestinal biopsies, interfering with the

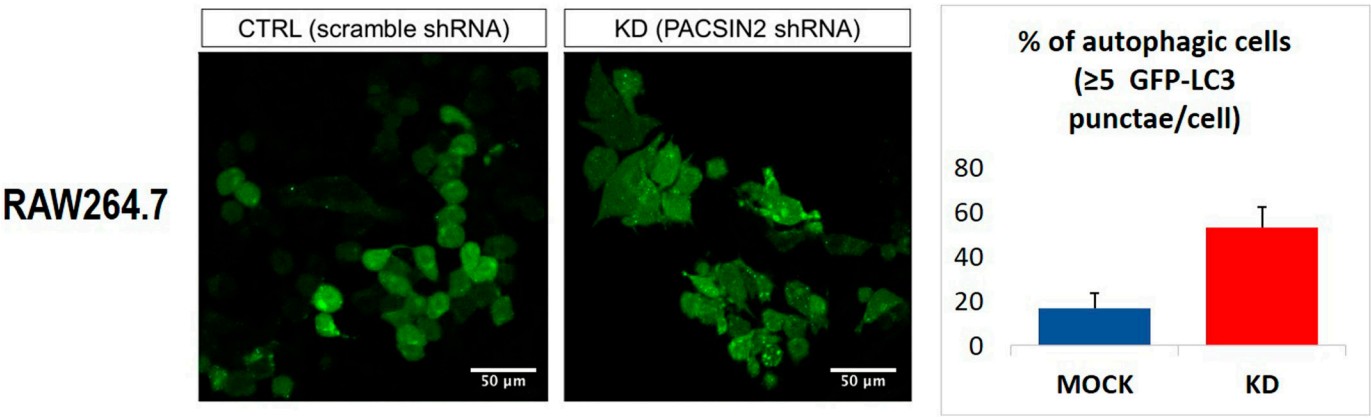

**Figure 4. GFP-LC3 *punctae* are increased after *PACSIN2* KD.**
Measurement of GFP-LC3 *punctae* was performed by confocal microscopy on confluent murine RAW 264.7 macrophages (n = 4, *t* test, *P* = 0.019). GFP-LC3 *punctae* were counted in 110 ± 45 control cells and in 115 ± 34 cells with *PACSIN2* KD.

intestinal biopsy results, *PACSIN2* levels were analyzed and compared in six samples of intestinal organoids and in six whole blood samples of IBD pediatric patients; no statistically significant differences in the *PACSIN2* amount were detected (Fig S5A). Consistently, similar PACSIN2 transcriptional levels were found analyzing intestinal biopsy cells and also specific intestinal biopsy–derived cell types, such as colon endothelium and fibroblast, compared with many cell types of the immunity system using the "IBD Transcriptome and Metatranscriptome Meta-Analysis" (IBD TaMMA) database (Zhu et al, 2018) (Fig S5B).

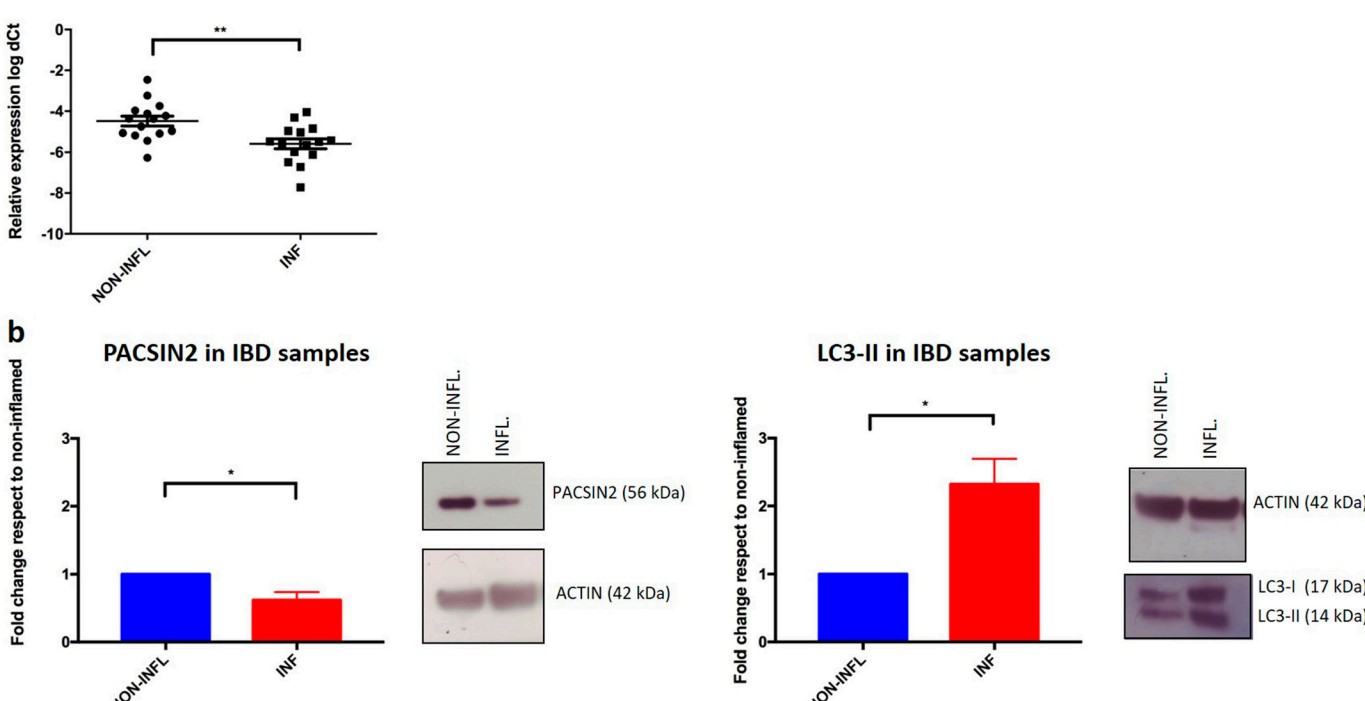

**Figure 5. PACSIN2 decreases in the presence of high autophagy and inflammation levels in colon biopsies of inflammatory bowel disease patients.**
**(A)** Comparison of *PACSIN2* gene expression level between samples of inflamed and non-inflamed colon biopsies on a cohort of 15 inflammatory bowel disease pediatric patients: *PACSIN2* levels were lower in inflamed colon tracts compared with non-inflamed ones (*t* test, *P* = 0.0084). **(B)** PACSIN2 protein expression level in patients of the same cohort is decreased (n = 6, *t* test, *P* = 0.017) in the presence of inflammation, corresponding to an increased LC3-II amount (n = 5, *t* test, *P* = 0.024). These results indicate an inverse correlation between PACSIN2 and autophagy levels, supporting the results observed in cell lines.

## PACSIN2 presents a LIR domain responsible for a protein–protein interaction with LC3

The examination of the primary structure for PACSIN2 protein (Fig S6) using the iLIR database[35] revealed the presence of one LIR domain (D-D-F-E-K-I), suggesting a potential physical interaction between PACSIN2 and LC3, a key protein in autophagosome formation and elongation. To determine whether PACSIN2 interacts with LC3, we performed co-immunoprecipitations between LC3 and PACSIN2 in NALM6 and LS180 cells under basal conditions. Immunoblotting analysis showed that PACSIN2 binds to LC3-II in both NALM6 (Fig 6A) and LS180 (Fig 6B) cells. In NALM6, experiments were performed also after chloroquine treatment to increase the available LC3 amount. These results confirmed the formation of a

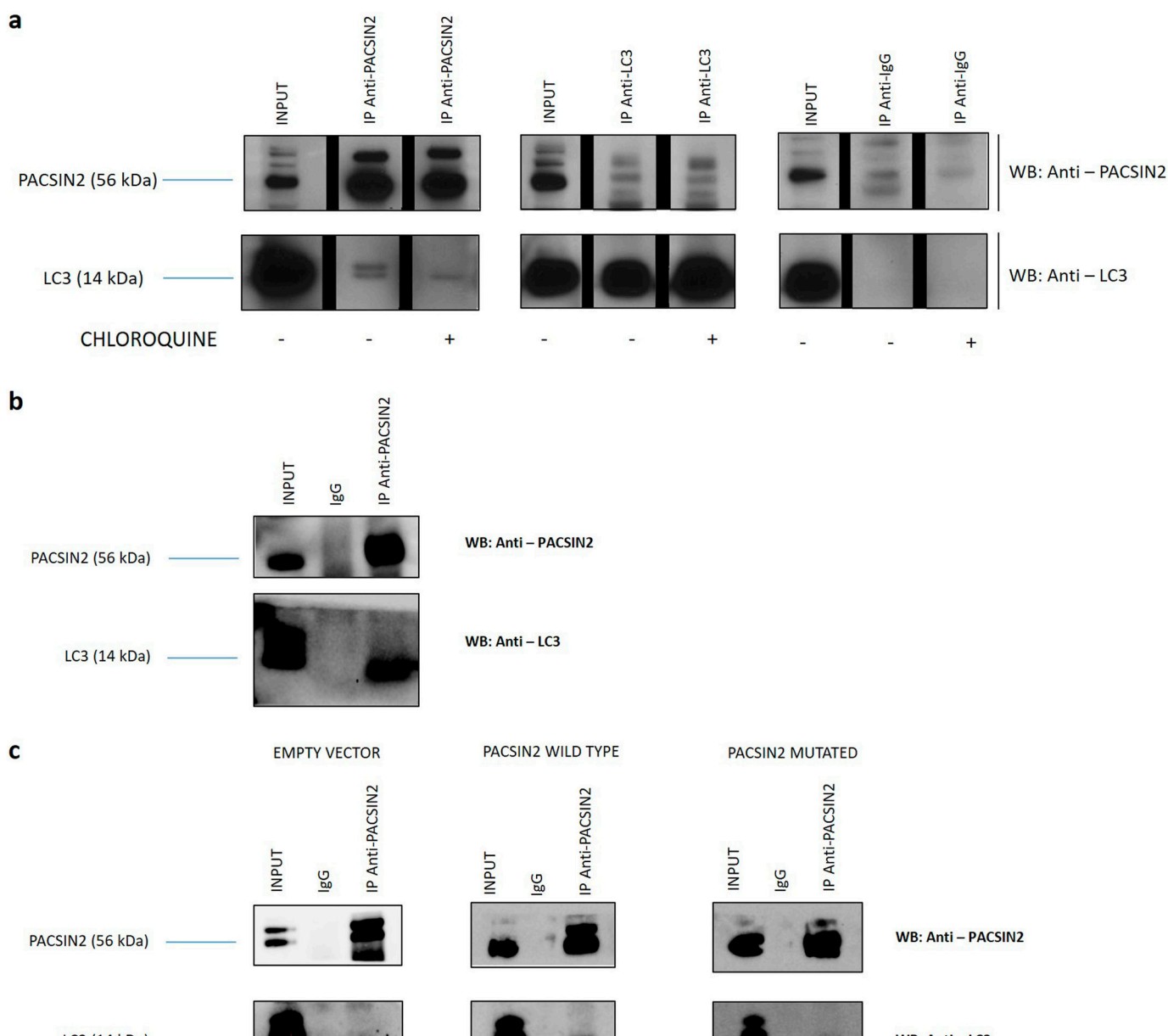

**Figure 6. PACSIN2 interacts with LC3.**
Co-immunoprecipitation (Co-IP) experiments performed with cellular lysates from NALM6 and LS180 cells (n = 3). PACSIN2 and LC3 were immunoprecipitated with anti-PACSIN2 or anti-LC3 specific antibodies. Co-immunoprecipitated proteins were detected by Western blot using the indicated antibodies. Co-IP performed with IgG was included as a control of specificity. INPUT is shown to assess the presence of the proteins of interest within the cellular lysates. **(A)** Co-IP with NALM6 lysates; the black lines in the blots indicated a splice between lanes. **(B)** Co-IP with LS180 lysates; **(C)** Co-IP with lysates obtained from LS180 cells transiently expressing wild-tpye PACSIN2 and mutated PACSIN2 (PACSIN2 protein with both the phenylalanine in position 109 and the isoleucine in position 112 replaced by alanine). Cells transfected with an empty expression vector are used as a control.
Source data are available for this figure.

complex between PACSIN2 and LC3, consistent with the hypothesis that PACSIN2 could play its role in the first phases of the autophagic flux.

To further demonstrate that the LIR domain of PACSIN2 is responsible for the protein–protein interaction between PACSIN2 and LC3-II, we performed the co-immunoprecipitations using LS180 transiently transfected with a plasmid coding for PACSIN2 wild-tpye isoform or one presenting PACSIN2 with a mutation in the LIR domain (both the phenylalanine in the third position of the LIR motif and the isoleucine in the last position of the LIR domain were substituted with alanine); cells transfected with an empty vector were used as a control. Results showed that by immunoprecipitating the mutated isoform of PACSIN2, the amount of co-immunoprecipitated LC3-II was lower (Fig 6C), confirming the hypothesis that the LIR motif plays a crucial role in the interaction between these two proteins.

### PACSIN2 KD impact on unfolded protein stress response evaluated as tunicamycin cytotoxicity

Autophagy could be used by cells as a survival mechanism in response to many different stress sources, such as ER stress; however, elevated stress levels lead to cell death. Because *PACSIN2* KD cell lines presented an up-regulated basal autophagy level, we tested the cytotoxic effects of tunicamycin, an ER stress inducer, that stimulates precipitation of unfolded protein aggregates, resulting in an autophagy stimulation. Interestingly, NALM6 KD cells presented an increased sensitivity to tunicamycin compared with control cells (−log IC50 NALM6 MOCK versus −log IC50 NALM6 KD: 6.794 ± 0.05 M versus 7.064 ± 0.06 M, *P* = 0.0279, Fig 7A). LS180 cells showed higher cytotoxicity to tunicamycin compared with NALM6 (−log IC50 NALM6 MOCK: 6.794 ± 0.05 M; −log IC50 LS180 MOCK: 8.3 ± 0.15 M); however, the presence of *PACSIN2* KD did not impact on tunicamycin cytotoxicity in LS180 cells (Fig 7B).

### PACSIN2 KD impact on mercaptopurine cytotoxicity

The role of PACSIN2 in mercaptopurine cytotoxicity was investigated in the presence of TPMT overexpression. Cytotoxicity assay results showed that NALM6 with the stable TPMT overexpression (NALM6*1, Figs S1 and S2) were the cells most sensitive to mercaptopurine treatment after 72 h of drug exposure and differed significantly from

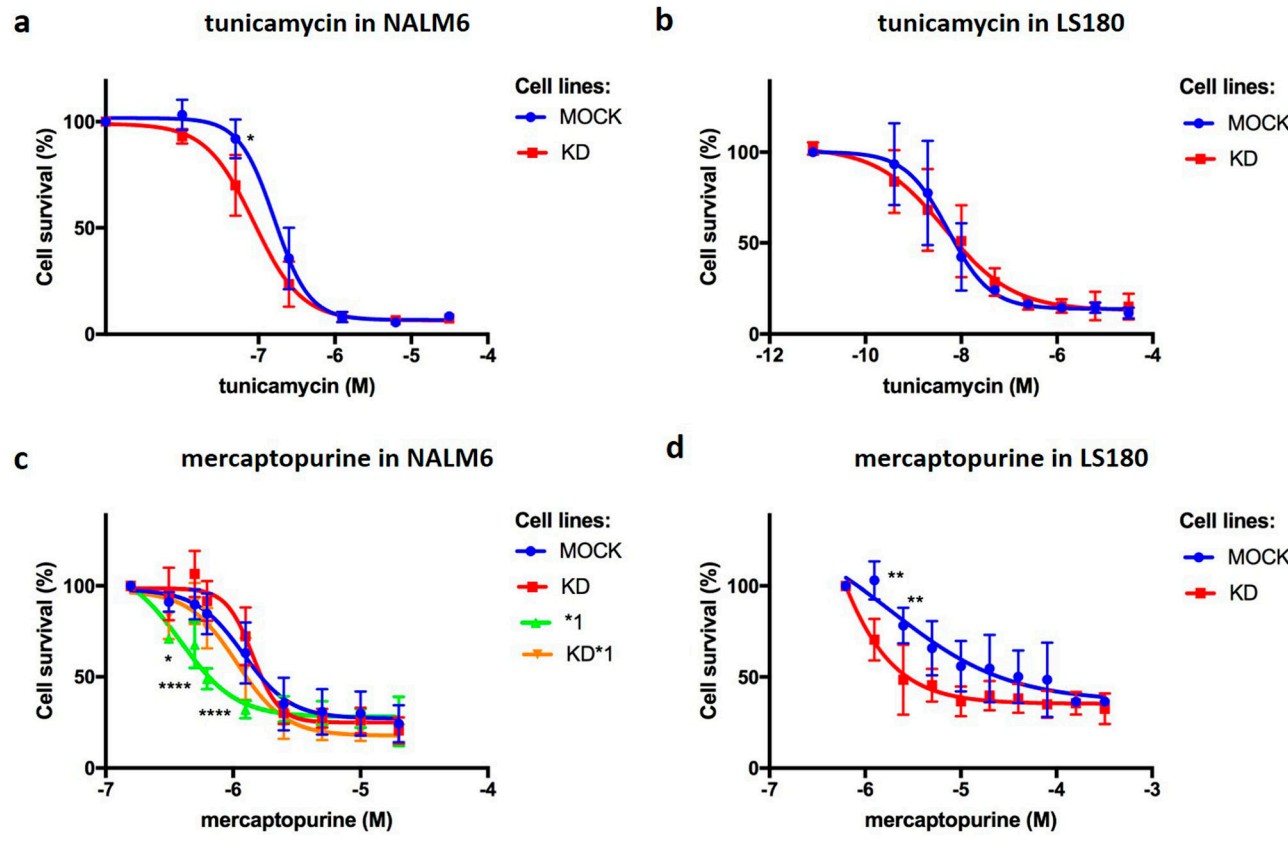

**Figure 7. Drug cytotoxicity assays.**
**(A)** NALM6 drug cytotoxicity assay to tunicamycin (−log IC50 NALM6 MOCK: 6.8 ± 0.05 M; NALM6 KD: 7.1 ± 0.06 M; NALM6 *1: 7.1 ± 0.12 M; NALM6 KD*1: 7.1 ± 0.11 M, n = 3, two-way ANOVA, *P* = 0.028) showed that NALM6 KD cells were more sensitive to this agent than control cells. No statistically significant difference in cytotoxicity was found in the presence of TPMT overexpression. **(B)** MTT assay result for tunicamycin cytotoxicity in LS180 cells showed no statistically significant difference in the presence of *PACSIN2* KD (−log IC50 LS180 MOCK: 8.3 ± 0.15 M; LS180 KD: 8.1 ± 0.21 M, n = 3). **(C)** NALM6 sensitivity assay for mercaptopurine: NALM6*1 was the most sensitive cell line to mercaptopurine treatment after 72 h and differed significantly from control cells (−log IC50 NALM6 *1 versus NALM6 MOCK: 6.4 ± 0.08 M versus 5.9 ± 0.04 M, n = 6, two-way ANOVA, *P* < 0.0001); interestingly, NALM6 KD*1 cells showed similar sensitivity to control cells, suggesting a role of PACSIN2 in TPMT activity in the presence of its overexpression. **(D)** LS180 drug cytotoxicity assay for mercaptopurine: LS180 KD cells were more sensible to mercaptopurine than control cells (−log IC50 LS180 MOCK: 5.3 ± 0.1 M; LS180 KD: 5.8 ± 0.04 M, n = 4, two-way ANOVA, *P* < 0.0001).

NALM6 MOCK (−log IC50 NALM6 *1 versus NALM6 MOCK: 6.4 ± 0.08 M versus 5.9 ± 0.04 M, *P* < 0.0001); no differences in mercaptopurine-induced cytotoxicity were detected between NALM6 MOCK and NALM6 KD cells. Interestingly, NALM6 with both stable *PACSIN2* KD and TPMT overexpression (NALM6 KD*1, Figs S1 and S2) showed similar sensitivity to NALM6 MOCK, suggesting a role of PACSIN2 in TPMT activity in the presence of its overexpression (Fig 7C). We tested the effect of *PACSIN2* KD on mercaptopurine cytotoxicity in LS180, and interestingly, LS180 KD cells were more sensitive to mercaptopurine than control cells (−log IC50 LS180 KD: 5.8 ± 0.04 M versus LS180 MOCK: 5.3 ± 0.1 M, *P* < 0.0001, Fig 7D), supporting the hypothesis of a possible tissue-specific role of PACSIN2 in intestinal cells.

### *PACSIN2* KD counteracts mercaptopurine-induced autophagy inhibition

On the basis of the increased cytotoxicity of mercaptopurine in the intestinal cell line LS180 with *PACSIN2* KD, we investigated the potential impact of mercaptopurine treatment on autophagy, using the drug concentrations displaying statistically different cytotoxicity (1.25 and 2.5 µM). In particular, the LS180 MOCK cells showed a higher percentage of survival compared with LS180 KD cells (103.08 ± 10.36 and 70.53± 11.48 for 1.25 µM mercaptopurine and 78.39 ± 9.8 and 48.53 ± 19.34 for 2.5 µM mercaptopurine in LS180 MOCK versus LS180 KD cells, respectively, Fig 7D). We evaluated the amount of the autophagic marker LC3-II after 24 or 48 h of mercaptopurine treatment; as a control, we used untreated samples.

After 24 h of mercaptopurine exposure, LC3-II was decreased in LS180 MOCK cells (fold change 1.25 µM: 0.72 ± 0.05, *P* = 0.0057; 2.5 µM: 0.59 ± 0.08, *P* = 0.0071, Fig 8A) and LS180 KD cells (fold change 1.25 µM: 0.85 ± 0.02, *P* = 0.0037; 2.5 µM: 0.84 ± 0.02, *P* = 0.048, Fig 8A) compared with untreated cells; 48 h of mercaptopurine treatment led to LC3-II reduction in both LS180 MOCK (fold change 1.25 µM MP: 0.53 ± 0.04, *P* = 0.0003; 2.5 µM MP: 0.14 ± 0.05, *P* < 0.0001, Fig 8B) and LS180 KD (fold change 1.25 µM MP: 0.71 ± 0.12, *P* = 0.023; 2.5 µM MP: 0.60 ± 0.04, *P* = 0.004, Fig 8B) cells, indicating that mercaptopurine is able to reduce LC3 amount in these cell lines.

Furthermore, LS180 KD cells presented higher LC3-II amount compared with MOCK cells after both 24 h (fold change untreated: 2.12 ± 0.06, *P* < 0.0001; 1.25 µM: 2.52 ± 0.3, *P* = 0.0004; 2.5 µM: 3.1 ± 0.2, *P* = 0.0007, Fig 8A) and 48 h (fold change untreated: 1.24 ± 0.07, *P* = 0.03; 1.25 µM: 1.62 ± 0.3, *P* = 0.035; 2.5 µM: 6.7 ± 2.35, *P* = 0.0015, Fig 8B), consistent with what was observed in confluent untreated cells (Fig 3B).

Taken together, these immunoblotting results indicate that mercaptopurine treatment reduces autophagy levels; interestingly, this effect is less evident in *PACSIN2* KD cells and is in line with the previous observations that PACSIN2 acts as a negative regulator of autophagy.

### Both mercaptopurine treatment and *PACSIN2* KD increase apoptosis in intestinal cells

We also investigated the potential impact of *PACSIN2* KD on apoptosis, after exposure of intestinal LS180 cells to mercaptopurine at 1.25 and 2.5 µM, evaluating the amount of cleaved PARP1 in intestinal LS180 cells by immunoblotting. LS180 KD cells showed increased apoptosis compared with control MOCK cells under basal condition after 24 h of cell growth (fold change 2.00 ± 0.26, *P* = 0.020, Fig 9A); interestingly, no statistically significant differences in cleaved PARP1 amount between LS180 MOCK and LS180 KD cells were detected after 48 h of cell growth without mercaptopurine treatment, indicating the presence of a similar apoptotic level in both cell lines at this time point. Treatment with 1.25 µM mercaptopurine for 24 h reduced apoptosis in both LS180 MOCK (fold change 0.32 ± 0.02, *P* < 0.0001) and LS180 KD (fold change 1.25 µM: 0.95 ± 0.07, *P* = 0.019) cells, compared with untreated controls, whereas exposure to 2.5 µM mercaptopurine increased apoptosis both in LS180 MOCK (fold change 5.21 ± 0.54, *P* = 0.0014, Fig 9A) and in LS180 KD (fold change 12.3 ± 0.51, *P* < 0.0001, Fig 9A) cells. Interestingly, consistently higher apoptosis was detected in LS180 KD cells exposed to mercaptopurine, compared with MOCK (fold change 1.25 µM: 2.97 ± 0.31, *P* = 0.001; 2.5 µM: 2.41 ± 0.2, *P* = 0.0007, Fig 9A) at 24 h. Furthermore, after 48 h of 2.5 µM mercaptopurine exposure, an increased apoptotic level was detected in LS180 KD cells (fold change 3.25 ± 0.42, *P* = 0.050) compared with the untreated samples, indicating higher apoptosis in LS180 KD cells at this time point.

Because the disruption of the mitochondrial membrane is one of the crucial steps in apoptosis activation by both intrinsic and extrinsic pathways, we performed the DiOC6 assay to evaluate the percentage of mitochondrial membrane polarization under basal condition and after 24, 48, and 72 h of mercaptopurine treatment; as a control, we used untreated cells cultured for the same time points. Results showed that after 72 h of mercaptopurine treatment, the percentage of the mitochondrial membrane potential was reduced in LS180 KD (untreated cells versus 1.25 µM: −15.95%, *P* = 0.026; untreated cells versus 2.5 µM: −26.42%, *P* < 0.0001), indicating increased apoptotic levels after mercaptopurine exposure. Interestingly, LS180 KD showed reduced mitochondrial membrane potential compared with MOCK control cells after 72 h of 2.5 µM mercaptopurine treatment (MOCK versus KD: −19.69%, *P* = 0.0012), further supporting increased apoptotic levels in LS180 KD cells after mercaptopurine treatment (Fig 9C).

Taken together with mercaptopurine cytotoxicity analyses in LS180 cells, these results suggest a possible contribution of apoptosis in mercaptopurine cellular death that is more evident in LS180 KD cells.

### PACSIN2 KD did not impact on the amount of thiopurine metabolites

On the basis of the cytotoxicity results in the intestinal LS180 cell lines, intracellular concentrations of thiopurine metabolites were evaluated through the HPLC-UV system after exposure of cells to 2.5 µM mercaptopurine for 24 and 48 h, to evaluate the effect of *PACSIN2* KD. *PACSIN2* KD did not have a significant impact on the amount of both thionucleotides (TGN) and methylated nucleotides (MMPN) (Fig S7). Taken together, these results suggest that *PACSIN2* KD does not affect thiopurine pharmacokinetics.

### PACSIN2 alters TPMT concentrations through a mechanism different from autophagy

Because PACSIN2 is capable of modulating the level of TPMT activity, we wanted to investigate whether autophagy could be the

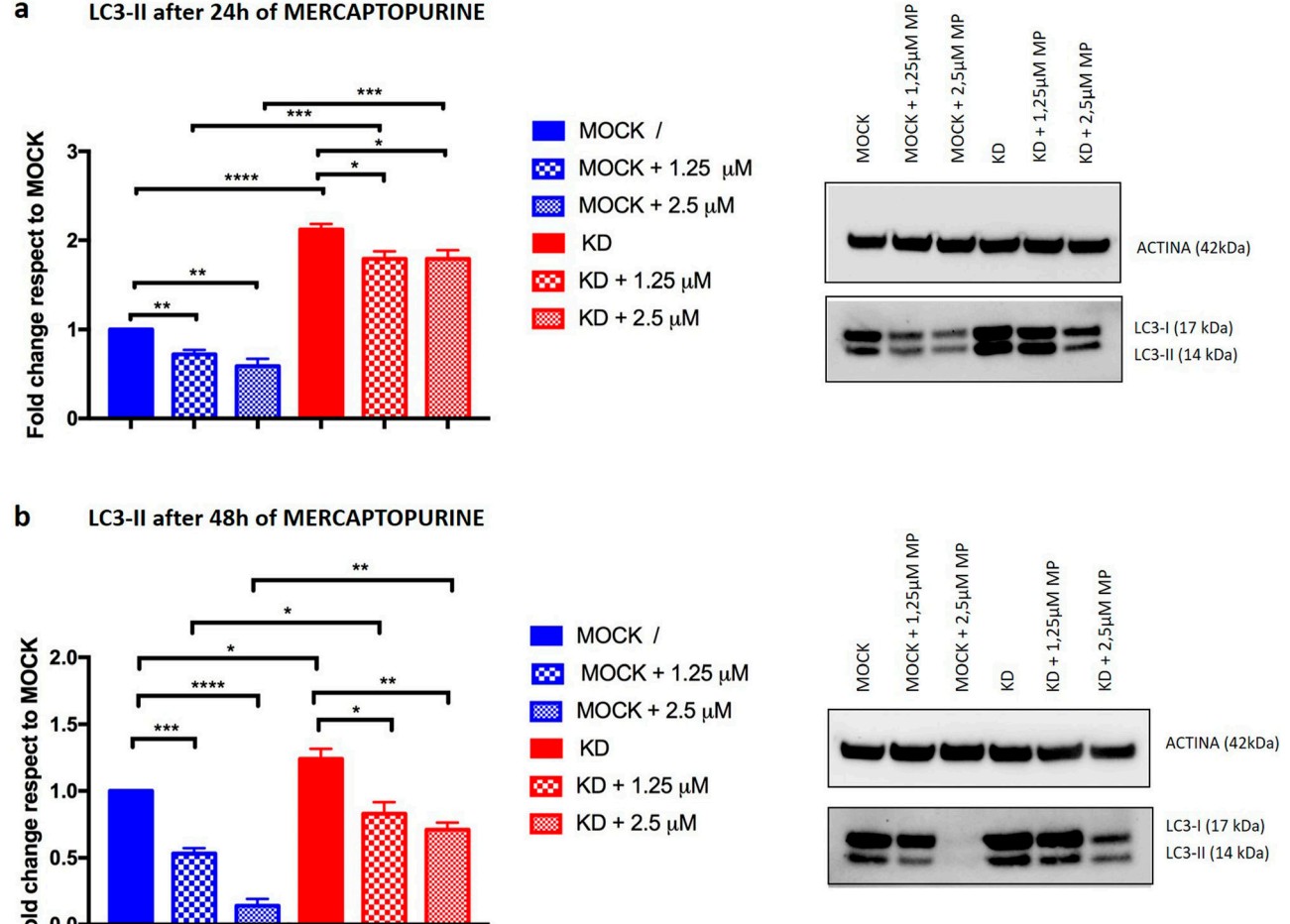

**Figure 8.  Mercaptopurine effects on autophagy.**
**(A)** Immunoblotting results of the autophagic marker LC3-II after 24 h of 1.25 and 2.5 $\mu M$ mercaptopurine treatment in LS180 cell lines. LS180 KD cells presented higher LC3-II levels compared with MOCK cells after 24 h (fold change untreated: 2.12 ± 0.06, n = 3, *t* test, *P* < 0.0001; 1.25 $\mu M$: 2.52 ± 0.3, n = 3, *t* test, *P* = 0.0004; 2.5 $\mu M$: 3.1 ± 0.2, n = 3, *t* test, *P* = 0.0007). Mercaptopurine exposure for 24 h reduced LC3-II levels in LS180 MOCK cells compared with untreated cells (fold change 1.25 $\mu M$: 0.72 ± 0.05, n = 3, *t* test, *P* = 0.0057; 2.5 $\mu M$: 0.59 ± 0.08, n = 3, *t* test, *P* = 0.0071) and LS180 KD cells (fold change 1.25 $\mu M$: 0.85 ± 0.02, n = 3, *t* test, *P* = 0.0037; 2.5 $\mu M$: 0.84 ± 0.02, n = 3, *t* test, *P* = 0.048). **(B)** LC3-II amount was reduced after 48 h of 1.25 $\mu M$ and 2.5 $\mu M$ mercaptopurine in LS180 cell lines. LS180 KD cells showed a higher amount of LC3-II compared with MOCK cells after 48 h (fold change untreated: 1.24 ± 0.07, n = 3, *t* test, *P* = 0.03; 1.25 $\mu M$: 1.62 ± 0.3, n = 3, *t* test, *P* = 0.035; 2.5 $\mu M$: 6.7 ± 2.35, n = 3, *t* test, *P* = 0.0015). Mercaptopurine treatment reduced LC3-II amount in both LS180 MOCK (fold change 1.25 $\mu M$ MP: 0.53 ± 0.04, n = 3, *t* test, *P* = 0.0003; 2.5 $\mu M$ MP: 0.14 ± 0.05, n = 3, *t* test, *P* < 0.0001) and LS180 KD (fold change 1.25 $\mu M$ MP: 0.71 ± 0.12, n = 3, *t* test, *P* = 0.023; 2.5 $\mu M$ MP: 0.60 ± 0.04, n = 3, *t* test, *P* = 0.004) cells, even if the decrease was less evident in LS180 KD cells. We performed immunoblotting using 10 $\mu g$ of cell lysate.

molecular mechanism on the basis of these observations. We first determined whether the turnover of TPMT*1 was increased in NALM6 KD*1 cells (Fig 10). Cells were incubated for different times with the mRNA translation inhibitor cycloheximide. The basal amount of PACSIN2 (t = 0) in NALM6 KD*1 (left panel) was 7% of that measured in NALM6 *1 control cells (right panel), indicating a complete KD of *PACSIN2* compared with control cells. The half-life of the residual amount of PACSIN2 in NALM6 KD*1 cells was decreased to 5 h compared with 39 h in NALM6*1 control cells. This is in accordance with immunoblotting and confocal microscopy results that showed an increased autophagy level in NALM6 KD cells. However, surprisingly, although the expression of TPMT at t = 0 in NALM6 KD*1 cells (left panel) was only 15% compared with NALM6*1 control cells (right panel), the half-life of TPMT was similar (43–59 h) in both cell lines. This indicates that the reduction of TPMT

expression in NALM6 KD*1 cells is unrelated to autophagy, suggesting that PACSIN2 alters cellular TPMT protein levels via a mechanism different from its effects on autophagy.

# Discussion

In this study, we demonstrate for the first time that PACSIN2 is an inhibitor of autophagy, acting at the beginning of the autophagic flux. PACSINs are proteins that functionally link actin cytoskeleton and vesicle formation by regulating tubulin polymerization, and exert their function mainly through protein–protein interactions with different substrates, such as N-WASP or dynamin. PACSINs have various biological roles, including modulation of endocytosis

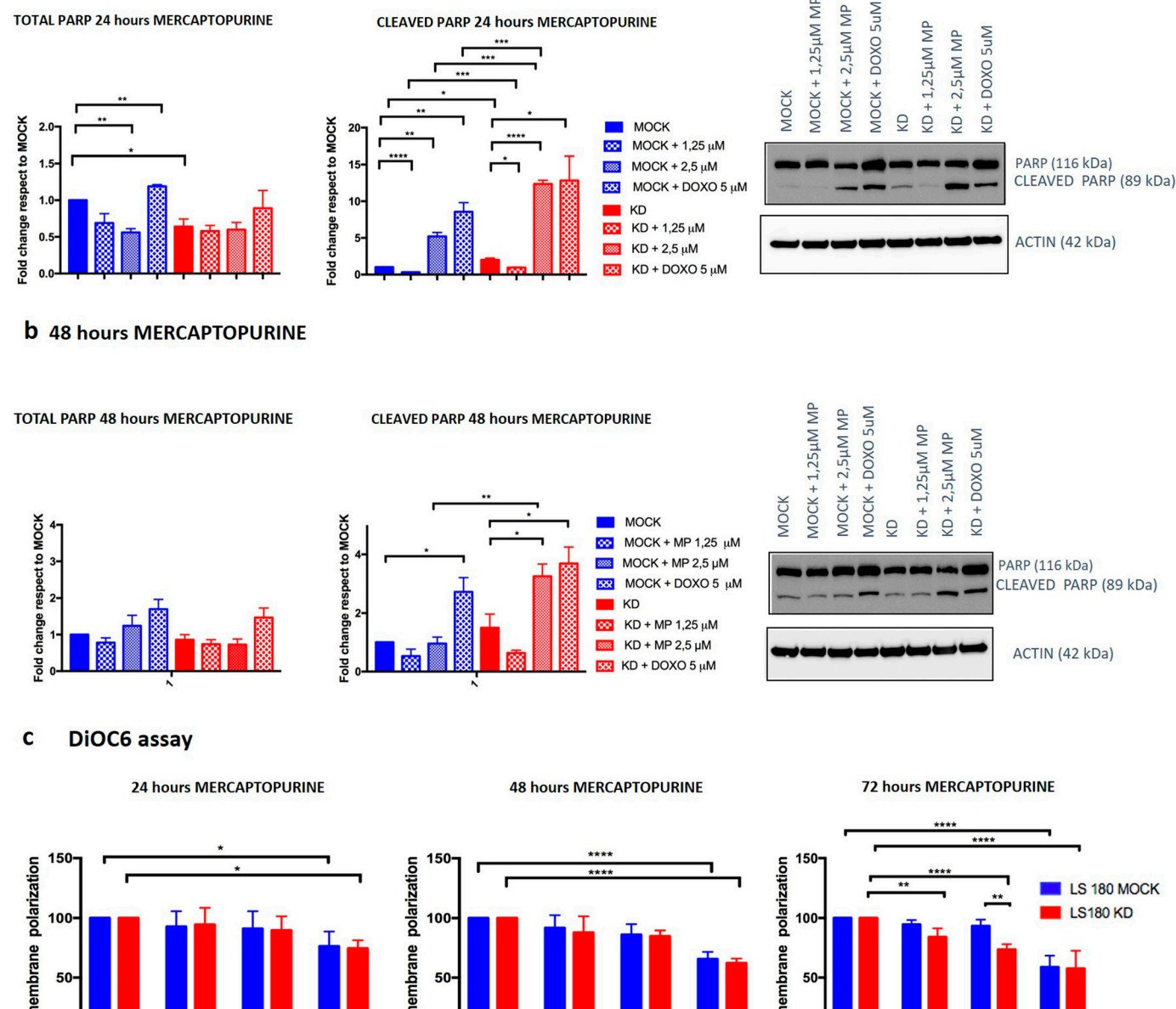

**a** 24 hours MERCAPTOPURINE

**b** 48 hours MERCAPTOPURINE

**c** DiOC6 assay

**Figure 9. Mercaptopurine effects on apoptosis.**
**(A)** Immunoblotting results of total and cleaved PARP1 in LS180 cell lines under basal condition or after 24 h of 1.25 $\mu$M and 2.5 $\mu$M mercaptopurine or 5 $\mu$M doxorubicin treatment in LS180 cell lines showed that drug-induced apoptosis increases in both cell lines; interestingly, this increase was higher in the presence of *PACSIN2* KD.
**(B)** Immunoblotting results of total and cleaved PARP1 in LS180 cell lines under basal condition or after 48 h of 1.25 $\mu$M and 2.5 $\mu$M mercaptopurine or 24 h of 5 $\mu$M doxorubicin treatment in LS180 cell lines showed that drug exposure induces apoptosis, especially in cells with *PACSIN2* KD. We performed immunoblotting using 10 $\mu$g of cell lysate. **(C)** DiOC6 assay results showed the percentage of the mitochondrial membrane polarization under basal condition and after 24, 48, or 72 h of mercaptopurine exposure; 24 h 5 $\mu$M doxorubicin treatment was used as a positive control. After treatment, cells with *PACSIN2* KD showed a lower mitochondrial membrane polarization, supporting the immunoblotting evidence about the potential role of mercaptopurine as an apoptotic inducer.

(de Kreuk et al, 2011; Dumont & Lehtonen, 2022) and membrane tubulation in cells such as megakaryocytes, likely contributing to platelet formation (Shimada et al, 2010; Dumont & Lehtonen, 2022).

PACSIN2 presents a high sequence homology with PACSIN1, which is implicated in signaling networks that control autophagy under optimal growth conditions (Szyniarowski et al, 2011; Oe et al, 2022).

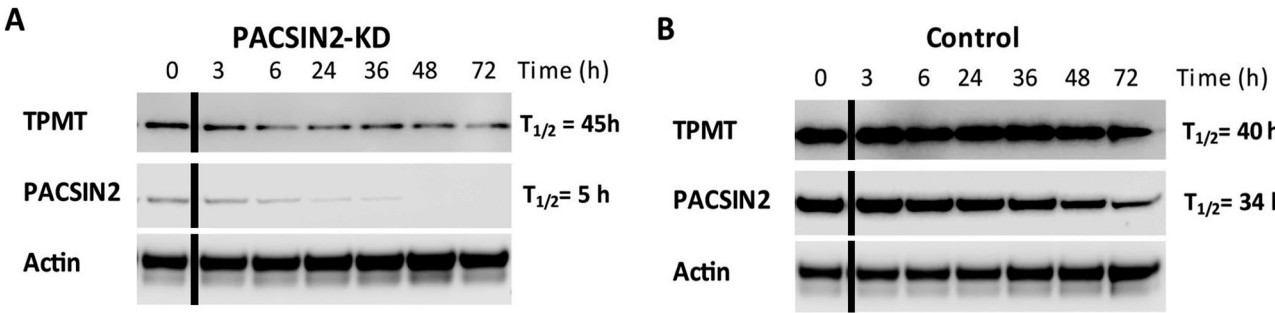

**Figure 10. TPMT\*1 protein stability is not influenced by *PACSIN2* KD.**
**(A)** Lysates of NALM6 cells overexpressing TPMT\*1 (NALM6\*1) with silenced *PACSIN2* (KD) or **(B)** control cells transduced with scramble shRNA (CONTROL) were incubated with the mRNA translation inhibitor cycloheximide and collected after 0, 3, 6, 24, 36, 48, and 72 h. 50 μg of cell lysates was analyzed on immunoblotting incubated with antibodies against PACSIN2, TPMT, and actin used as a loading control. The experiment was repeated two times obtaining consistent results. The black lines in the blots indicated a splice between lanes.
Source data are available for this figure.

To elucidate PACSIN2 involvement in autophagy, we followed the paradigm that autophagy-related proteins, such as SQSTM1/P62 (Mathew et al, 2009) or ATG16L1 (Cadwell et al, 2008), are accumulated in autophagy-deficient cells and are reduced after induction of autophagy by stimuli such as rapamycin. Our results showed that in autophagy-deficient Atg7$^{-/-}$ MEF cells, PACSIN2 protein amount was about two times higher than in wild-tpye control MEF cells; moreover, after induction of autophagy by rapamycin, PACSIN2 concentration decreased only in autophagy-proficient Atg7$^{+/+}$ MEF cells, indicating a possible involvement of PACSIN2 in the autophagy machinery. Similarly, SQSTM1/P62 resulted accumulated in immortalized baby mouse kidney cells presenting *Beclin1* or *Atg5* knockout and therefore autophagy-deficient MEFs (Mathew et al, 2009), whereas ATG16L1 concentration was reduced after rapamycin exposure in autophagy-proficient MEFs (Cadwell et al, 2008). To investigate further the role of PACSIN2 in autophagy, we evaluated the effect of *PACSIN2* KD in different in vitro models representative of the lymphoid and intestinal tissues on the autophagic markers LC3 and SQSTM1/P62. LC3 is a key autophagy protein involved in elongation of the phagophore membrane and in autophagosome maturation; upon induction of autophagy, cytosolic LC3-I is converted to LC3-II by conjugation to phosphatidylethanolamine and subsequently translocated to autophagosomal membranes (Kabeya et al, 2004).

Overall, increased autophagy was observed when *PACSIN2* was silenced, suggesting that this protein could be a negative modulator of autophagy. In particular, higher LC3-II levels and lower SQSTM1/P62 were detected after *PACSIN2* KD in both cell lines; consistently, recent results on HeLa cells showed higher LC3-II levels after *PACSIN1* KO, indicating a role of PACSIN1 as a negative modulator of autophagy (Oe et al, 2022); moreover, a similar approach was used in a previous study on osteosarcoma U2OS cells and MEF cells transfected with siRNA for silencing *BAP31*, an ER membrane protein acting as autophagic negative modulator, that showed a higher amount of LC3-II protein expression level and LC3-GFP *punctae* (Machihara & Namba, 2019). The human protein Atlas (Human Protein Atlas proteinatlas.org) (Uhlén et al, 2015) showed a higher LC3 expression in intestine compared with lymphoid tissue, which could be explained by a different epigenetic control (Lee &

Lee, 2016). Consistently, our immunoblotting results for LC3 indicated a higher autophagy basal level in LS180 cells compared with NALM6. Interestingly, a higher LC3-I amount after *PACSIN2* KD was detected in NALM6 cells and not in LS180 cells. The higher LC3-I conversion to LC3-II in NALM6 KD cells because of autophagosome formation might stimulate the transcription of LC3, leading to a statistically significant increased level of LC3-I (Zhu et al, 2018; Di Malta et al, 2019).

In the intestine, autophagy plays a role in the prevention of inflammation and altered autophagy levels have been associated with IBD (Lapaquette et al, 2015; Shao et al, 2021). Interestingly, both PACSIN2 gene expression and protein concentrations were decreased in the presence of higher autophagy levels in inflamed colon samples from a cohort of pediatric IBD patients, supporting the inverse correlation between PACSIN2 amount and autophagy levels. Consistently, a recent study identified an increased amount of the autophagy protein Atg10 in colon biopsies of IBD patients compared with healthy donors (Abbasi Teshnizi et al, 2021). The presence of disrupted autophagy is considered a contributing factor for IBD pathogenesis (Iida et al, 2017), and also, different polymorphisms in autophagy genes have been associated with pediatric IBD onset through genome-wide association studies (Khor et al, 2011; Iida et al, 2017). From a clinical point of view, the drug-mediated autophagy modulation is considered a promising mechanism for IBD clinical management, and different drugs, involved in autophagy modulation, are currently under clinical trial (Azzman, 2019; Retnakumar & Muller, 2019). Because PACSIN2 is a negative autophagy modulator, it would be interesting to investigate whether PACSIN2 targeting and modulation by a small molecule could affect autophagy in intestinal models and result beneficial for IBD patients. In order to understand the mechanisms by which *PACSIN2* KD causes LC3-II to increase and identify the step of autophagic flux where PACSIN2 is involved, we blocked the last phases of the autophagic process by chloroquine treatment (Jiang & Mizushima, 2015; Yoshii & Mizushima, 2017). A further increase in autophagy in *PACSIN2* KD cells treated with chloroquine was documented, suggesting that PACSIN2 impairs the autophagic flux during the early phases and does not cause a defect in lysosomal degradation (Yoshii & Mizushima, 2017). A similar result was

observed by Machihara and collaborators that exposed U2OS cells knocked out for *BAP31* to bafilomycin A1, a substance with the same mechanism of action of chloroquine (i.e., autophagosome–lysosome fusion blockage (Fedele & Proud, 2020)) bafilomycin A1 treatment, increased LC3-II particularly in cells without *BAP31* (Machihara & Namba, 2019).

Different autophagy modulators interact with LC3 (Mizushima et al, 2010; Birgisdottir et al, 2013; Chun & Kim, 2018). PACSIN2 protein shows different interacting region domains, responsible for a physical interaction with other proteins, such as Rac-1, Cobll1, and SH3BP1, that modulate different cellular biological processes and impacts on tyrosine kinase inhibitor sensitivity in *in vitro* models (de Kreuk et al, 2011; Park et al, 2022). Interestingly, we found that PACSIN2 presents a LIR domain in its N-terminal amino acid sequence, responsible for a physical interaction with LC3 that may permit to exert PACSIN2 regulation on autophagy. Consistently, Pankiv et al (2007) identified a LIR motif on SQSTM1/P62 primary protein sequence and demonstrated that it is fundamental for the binding to LC3. In particular, the binding between SQSTM1/P62 and LC3 is mediated by a 22-residue sequence in the N-terminal region of the SQSTM1/P62 protein and contains an evolutionarily conserved motif (D-D-W-T-H-L), presenting an aromatic amino acid in the central position and an aliphatic amino acid in the last position of the motif, similar to the LIR motif of PACSIN2, that presents amino acids with the same characteristics in the third and in the last position of the domain (Pankiv et al, 2007). Many different autophagic receptors, such as NBR1, FYCO1, NIX, PLEKHM1, and JIP1, involved in the autophagy regulation, present a LIR domain that represents the fundamental molecular mechanisms through which these proteins modulate autophagy (Novak et al, 2010; Rozenknop et al, 2011; Fu et al, 2014; Olsvik et al, 2015; McEwan et al, 2015).

Autophagy could be used by cells as a survival mechanism in response to many different stress sources, such as ER stress (Ogata et al, 2006); however, in elevated stress conditions, autophagy may contribute to cell death (Liu et al, 2017). The cytotoxicity assay results of tunicamycin showed a higher cytotoxic effect of this compound on LS180 cells compared with NALM6 cells; consistently, the intestinal LS180 cells presented a higher basal autophagy level compared with NALM6, and the exposure to tunicamycin, which increases cellular stress, might induce a further cell death both in LS180 MOCK and in KD cells, not permitting to detect the possible effect of *PACSIN2* KD in this intestinal model. The effect of *PACSIN2* KD after tunicamycin exposure was observed in NALM6 cells: NALM6 KD cells presented an increased sensitivity to the ER stress inducer tunicamycin compared with control cells, probably because the loss of *PACSIN2* increased autophagy and the cell exposure to a further stress source, such as tunicamycin treatment, affected in a significant way the cell vitality. The presence of unfolded proteins stimulates ER stress response, activating the unfolded protein response pathway, leading to the induction of mTORC1, an autophagy negative regulator that stimulates SQSTM1/P62 accumulation (Guha et al, 2017). A previous study performed on wild-type (SQSTM1/P62$^{+/+}$) and SQSTM1/P62-deficient (SQSTM1/P62$^{-/-}$) MEF cells treated with tunicamycin found that the absence of SQSTM1/P62 stimulates apoptotic cell death (Park et al, 2016). Consistently, our results showed that NALM6 KD cells have higher autophagy levels, associated with decreased SQSTM1/P62 amount and increased tunicamycin sensitivity.

Our observation that TPMT protein concentrations were reduced in *PACSIN2* KD cells is concordant with our previous reports that PACSIN2 modulates TPMT activity in patients with ALL (Stocco et al, 2012). Indeed, we observed also that TPMT overexpression in NALM6 cells increased mercaptopurine sensitivity, as described before. In particular, this higher sensitivity in NALM6*1 cells compared with MOCK is due to the supra-physiological accumulation of methylated metabolites, which occurs *in vitro* upon TPMT overexpression (Dervieux et al, 2001). Interestingly, the presence of *PACSIN2* KD was able to rescue this effect, supporting the influence of PACSIN2 on TPMT activity, previously demonstrated both in these cell lines and in ALL pediatric patients (Stocco et al, 2012). However, although *PACSIN2* KD consistently lowers the cellular concentration of TPMT*1 protein, the degradation of TPMT*1 is not affected by the induction of autophagy. This result is concordant with previous reports, which showed that TPMT is targeted for degradation by the proteasome rather than autophagy (Tai et al, 1999; Li et al, 2008). In contrast, the half-life of residual PACSIN2 is greatly reduced after its KD, likely because of the demonstrated increased autophagy in *PACSIN2* KD. Interestingly, the HPLC-UV analysis showed that *PACSIN2* KD did not affect concentraions of thiopurine metabolites, indicating that PACSIN2 is not able to affect thiopurines pharmacokinetics, but rather their pharmacodynamics.

We previously reported that *PACSIN2* variants increased the severity of gastrointestinal toxicity during consolidation therapy for ALL (Stocco et al, 2012). Interestingly, *PACSIN2* KD increased significantly mercaptopurine sensitivity in intestinal cell lines but not in leukemia cell lines, suggesting a tissue-specific effect of PACSIN2 in the intestine; however, further investigations are needed to shed light on the molecular mechanisms on the basis of the biological function of PACSIN2 in different tissues. PACSIN2 was also reported to interact directly with Rac-1, a protein whose inhibition by thiopurine's active metabolites has been involved in thiopurine's mechanism of cytotoxicity (de Kreuk et al, 2011). Therefore, the interaction between PACSIN2 effects on autophagy and Rac-1 inhibition by thiopurines may govern thiopurine effects on cell lines exposed to these drugs and should be further explored. To investigate the possible tissue-specific role of PACSIN2 in the intestine, we evaluated autophagy and apoptosis levels after mercaptopurine treatment and found that mercaptopurine increased apoptosis and, surprisingly, reduced autophagy in LS180 intestinal cells. This result on autophagy is in contrast with a previous study performed on other stabilized intestinal cell lines (HT29 cells), which found increased LC3-II levels after mercaptopurine exposure. However, Chaabane and collaborators evaluated autophagy after cell exposure to higher concentrations of mercaptopurine (50 μM mercaptopurine for 24 h) compared with the drug concentrations we used (1.25 and 2.5 μM) in our treatment protocol: indeed, after exposure to mercaptopurine, cells were grown without treatment for 72 h and then analyzed (Chaabane & Appell, 2016). The replacement of medium with drug with a fresh medium could be a source of energy that may stimulate the activation of survival mechanisms, such as autophagy. From a clinical point of view, the mercaptopurine concentrations used in the current study were more therapeutically relevant compared with those used in Chaabane's study; moreover, the intestinal *in vitro* model LS180 used in the current study is commonly considered a good model to investigate the effects of the intestinal drug

metabolism and to test the cytotoxic effect of drugs (van de Kerkhof et al, 2007).

Autophagy can induce or block apoptosis depending on the type of cell, nature, and duration of the stimulus (Xie et al, 2020). Molecular mechanisms on the basis of the cross-talk between the two pathways remain unclear because the interconnection between autophagy and apoptosis may occur at different stages and may involve different molecules such as Beclin1, Bcl-2, PINK1, and SQSTM1/P62 (Djavaheri-Mergny et al, 2010; Zhang et al, 2018; Wang et al, 2019; Zhu et al, 2021). Our results showed that in the presence of mercaptopurine, the higher autophagy levels because of the lack of PACSIN2 promote apoptosis, leading to increased drug sensitivity in LS180 KD cells compared with LS180 MOCK. The direct correlation between autophagy and apoptotic cell death had been already reported in literature for other compounds and in vitro models. It is difficult to identify the specific molecular mechanism of PACSIN2 in the interplay between autophagy and apoptosis, but we hypotesize that SQSTM1/P62 could be involved in this interplay, because it has been demonstrated that the increased amount of this protein in the presence of lower autophagy levels leads to apoptosis induction (Zhu et al, 2021).

Although the results about the inverse correlation between PACSIN2 and autophagy levels are encouraging and consistent between cell lines and intestinal biopsies, the evaluation only ex vivo of the association and in vitro of the causality between PACSIN2 amount and autophagy levels and the lack of in vivo assessment of the causality represent limitations of the study and future experiments need to be planned accordingly. Another limitation of the current study is that although we demonstrated that PACSIN2 modulates both autophagy and TPMT protein concentration, we did not observe an influence on TPMT protein degradation. Therefore, other molecular mechanisms must be considered, such as direct modulation of TPMT mRNA concentration by PACSIN2, consistent with our previous studies showing that PACSIN2 KD is associated with reduced TPMT protein concentration and TPMT mRNA levels (Stocco et al, 2012). This hypothesis needs to be evaluated by further studies.

Taken together, our findings demonstrate for the first time a clear role of PACSIN2 as an inhibitor of autophagy, putatively through inhibition of autophagosome formation by a protein–protein interaction with LC3-II mediated by a LIR motif present in the PACSIN2 sequence. Moreover, we provide in vitro evidences of the role of PACSIN2 as a modulator of mercaptopurine-induced cytotoxicity in intestinal cells, suggesting that PACSIN2-regulated autophagy might influence thiopurine sensitivity. Prospectively, these findings could be relevant for the personalization of therapy in pediatric patients needing thiopurines.

# Materials and Methods

## Cell lines and treatments

The following stabilized cell lines were used: (I) MEFs, both as Atg7 wild-type (Atg7$^{+/+}$) autophagy-competent control cells and as Atg7 knockout cells (Atg7$^{-/-}$), used as autophagy-deficient models (Taherbhoy et al, 2011); (II) human peripheral blood leukemia pre–B cells NALM6; (III) NALM6 presenting the forced TPMT*1

overexpression (NALM6*1); (IV) human colon adenocarcinoma cells LS180; (V) human epithelial cervix adenocarcinoma HeLa cells with forced expression of GFP-LC3 (HeLa$^{GFP-LC3}$); and (VI) murine macrophages RAW 264.7 with forced expression of GFP-LC3 (RAW$^{GFP-LC3}$). NALM6, NALM6*1, LS180, HeLa$^{GFP-LC3}$, and RAW$^{GFP-LC3}$ cell lines were stably engineered to knock down PACSIN2 (KD); as a control, we used the same cell line transfected with a scramble vector (MOCK), as previously described (Stocco et al, 2012, Figs S1 and S2). MEF, HeLa$^{GFP-LC3}$, and RAW$^{GFP-LC3}$ cell lines were grown in DMEM containing 10% FBS and 20 mM glutamine; NALM6 and LS180 cell lines were grown in RPMI 1640 containing 10% FBS and 20 mM glutamine. Cells were cultured and maintained according to standard procedures.

PACSIN2 protein stability was investigated after autophagy induction by rapamycin and after inhibition of protein synthesis by cycloheximide exposure (Cadwell et al, 2008). The autophagic flux was evaluated by exposing cells to chloroquine, which impairs autophagosomes fusion with lysosomes (Yoshii & Mizushima, 2017). The effect of PACSIN2 KD on unfolded protein response induced by tunicamycin exposure was investigated (Park et al, 2016). For mercaptopurine, whose toxicity on intestinal and lymphoid cells has been associated with PACSIN2 variant (Stocco et al, 2012), we evaluated the effect of PACSIN2 KD on cytotoxicity, considering also induction of autophagy and apoptosis. Furthermore, cells were exposed to cycloheximide alone, to evaluate the effect of PACSIN2 KD on TPMT protein stability (Kao et al, 2015). Treatments used to evaluate the impact of PACSIN2 KD on autophagy and mercaptopurine sensitivity in different cell lines are summarized in Table S1.

## Patient samples

Thirtytwo pediatric patients (mean age: 13.9 ± 0.74 years, 17 males [53.12%]; 19 of them presented Crohn's disease [59.4%], whereas 13 were affected by ulcerative colitis [40.6%]) were enrolled at diagnosis at the Gastroenterology Unit of the Pediatric Department of the Institute for Maternal and Child Health IRCCS Burlo Garofolo in Trieste, Italy. Local ethical committee approval for the study was obtained. The study was conducted in accordance with the principles outlined in the Declaration of Helsinki, and the parents of all the participating children gave written informed consent before study participation. Biopsies were collected in 26 patients, during a diagnostic colonoscopy. Inflamed and non-inflamed tissues of 15 patients were collected in TRIzol (15596026; Thermo Fisher Scientific) for RNA isolation, whereas inflamed and non-inflamed tissues from other five patients were used for protein lysate preparation. For six IBD pediatric patients, colon biopsies were used to generate intestinal organoids: crypts embedded in Matrigel (CLS356231; Corning) were cultured and passaged as described by Jung and colleagues (Jung et al, 2011). From this intestinal model, the RNA was extracted and used for the gene expression analyses. Finally, for six patients the RNA deriving from whole blood was available for the gene expression analyses.

## Immunoblotting

Cells were pelleted by 5-min centrifugation at room temperature at 300g, washed once with PBS, and lysed as follows. MEF and HeLa$^{GFP-LC3}$

cell lines were lysed with RIPA buffer (50 mM Tris–HCl, 150 mM NaCl, 1.0% NP-40, 0.5% sodium deoxycholate, 1 mM EDTA, 0.1% SDS, and 0.01% sodium azide) supplemented with Complete Protease Inhibitor Cocktail (04693116001; Roche). NALM6 and LS180 cell lines were lysed in a buffer containing 10 mM Tris–HCl, pH 7.4, 0.1% SDS, 100 mM NaCl, and 100 mM EDTA, supplemented with Halt Protease Inhibitor Cocktail 1× (87786; Thermo Fisher Scientific). Protein quantification was performed by the Bradford analysis (B6916; Sigma-Aldrich), and absorbance was measured at 570 nm on a Microplate Reader EL311 (214891; BioTek Instruments, Inc.). Ten biopsies (five non-inflamed and five inflamed) of the intestinal tracts of pediatric patients with IBD were lysed in RIPA buffer supplemented with Halt Protease Inhibitor Cocktail 1× (87786; Thermo Fisher Scientific). Protein quantification was performed by Pierce BCA Protein Assay Kit (23227; Thermo Fisher Scientific), and absorbance was measured at 570 nm. Equal amounts of proteins (10–50 µg, as specified in the immunoblotting results in figure legends) were separated on NuPAGE 10% Bis-Tris protein gels (NP0301BOX; Life Technologies) or Bolt 4–12%, Bis-Tris, 1.0 mm Mini Protein Gel (NW04122BOX; Thermo Fisher Scientific). Proteins were then transblotted to nitrocellulose membranes (PB7320; Thermo Fisher Scientific) using Electrophoresis Power Supply (EPS301; Thermo Fisher Scientific). After incubation with 5% non-fat milk in Tris-buffered saline (50 mM Tris-Cl and 150 mM NaCl, pH 7,5) with 0.1% Tween-20 (T-TBS) for 1 h, membranes were incubated with rabbit primary antibodies against human proteins LC3 (1:1,000 dilution; ab48394; Abcam), PACSIN2 (1:100 dilution; TA325022; OriGene), GAPDH (1:500 dilution; sc20357; Santa Cruz), actin (1:3,000 dilution; ab218787; Abcam), SQSTM1/P62 (1:250 dilution; GT381; GeneTex), PARP1 (1:1,000 dilution; #9532; Cell Signaling Technology), or mouse primary antibody against human TPMT (1:500 dilution; sc-374154; Santa Cruz), followed by incubation with appropriate HRP-conjugated secondary anti-rabbit IgG (1:10,000 dilution; AP132P; Merck) or HRP-conjugated secondary anti-mouse IgG (1:20,000 dilution; anti-mouse IgG, HRP-linked Antibody #7076; Cell Signaling Technology). Immunocomplexes were visualized by chemiluminescence using the LiteAblot TURBO Extra Sensitive Chemiluminescent Substrate (EMP012001; Euroclone) and using photographic films (Carestream Kodak Biomax, Z373508; Merck) or with the ChemiDoc gel imaging system (Bio-Rad). Protein levels were quantified by densitometry normalized against GAPDH or actin. Signal intensities were quantified using ImageJ and normalized for the signal intensity of the loading control in the same lane.

## Confocal live imaging

Because of the difficulty of performing imaging experiments in suspension cells such as NALM6, imaging experiments were performed in adherent HeLa[GFP-LC3] and RAW 264.7[GFP-LC3] cell lines to measure GFP-LC3 *punctae*, considered as a gold standard assay to assess autophagosome quantity in cells (Kabeya et al, 2004; Klionsky et al, 2014). The number of autophagic cells was measured by the percentage of cells containing five or more GFP-LC3 *punctae* in the same confocal field. Live-cell imaging was performed with the inverted microscope Nikon TE2000-E equipped with a confocal system C1Si (Nikon), an argon laser at 488 nm, and DPSS lasers at

404 and 561 nm (Melles Griot). Temperature was kept at 37°C and $CO_2$ at 5% using an environmental control chamber (In Vivo Scientific). Images were acquired on cells cultivated at 70% confluence, using a 40× objective with oil immersion.

## Gene expression analysis on patients' biopsies

Total RNA was extracted from patients' biopsies using TRIzol reagent (15596026; Thermo Fisher Scientific) according to the manufacturer's instructions; RNA concentration and purity were evaluated using NanoDrop instrument (NanoDrop 2000; Thermo Fisher Scientific). The reverse transcription reaction was performed using the High Capacity RNA-to-cDNA Kit (4387406; Thermo Fisher Scientific), and quantitative real-time (RT) PCR was carried out in duplicate using the TaqMan Gene Expression Assay to assess *PACSIN2* mRNA expression, according to the manufacturer's instructions. *PACSIN2* expression values were normalized using the ribosomal protein lateral stalk subunit P0 (*RPLP0*) as a reference gene for the intestinal biopsy samples and using actin for the samples deriving from intestinal organoids and the whole blood. Results were expressed as base 10 logarithm of the ΔCt of the relative expression (RE).

## LIR identification

The examination of the primary structure for PACSIN2 protein to identify a possible LIR domain in the sequence was performed through an in silico analysis performed using the iLIR database (Jacomin et al, 2016). As a positive control, we performed the same analysis on SQSTM1/P62 primary sequence to identify the LIR domain present in this protein (Pankiv et al, 2007); finally, this in silico analysis was performed on the ABL1 primary sequence, used as a negative control.

## Transient cell transfection

OmicsLink Expression-Ready ORF cDNA Vector bearing only the eGFP reporter gene (Cat# EX-EGFP-M39), and vector carrying the *PACSIN2* wild-type and mutated cDNA (Cat# CS-I2075-M39 and CS-I2075-M39-01, respectively) were purchased from GeneCopoeia. *PACSIN2* wild-type cDNA sequence (NM_007229.3) encoded for the protein sequence is shown in Fig S4. For mutated *PACSIN2*, the nucleotide sequence TTCGAGAAGATC in position 325 of the NCBI Reference Sequence: NM_007229.3N was mutated into GCAGA-GAAGGCA resulting in a PACSIN2 protein with both the phenylalanine in position 109 and the isoleucine in position 112 replaced by alanine. For cell line transient transfections, plasmids were expanded using DH5α *E. coli*–competent cells grown in Luria–Bertani broth (NaCl 10 g/l, tryptone 10 g/l, and yeast extract 5 g/l) and purified using the EndoFree Plasmid Maxi Kit (12362; Qiagen). Plasmid quantification was performed using NanoDrop instrument (NanoDrop 2000; Thermo Fisher Scientific). LS180 MOCK cells were seeded in RPMI 1640 without antibiotics at a concentration of $1 × 10^6$ cells/well; cells at 80–90% confluence were transfected with 7 µg of plasmid using 6 µl of Lipofectamine 2000 Transfection Reagent (11668019; Thermo Fisher Scientific) and 2 ml of Opti-MEM (31985070; Thermo Fisher Scientific) and incubated for 48 h.

## Co-immunoprecipitations

Co-immunoprecipitations were performed using NALM6 cells under basal condition and after 3 h of 50 $\mu M$ chloroquine exposure, thus resulting in an increased amount of available LC3. Moreover, we performed co-immunoprecipitations in LS180 MOCK cells transfected to overexpress the wild-type and mutated PACSIN2. Cells were grown for 48 h at 37°C and 5% $CO_2$, pelleted by centrifugation, washed once with PBS, and lysed using a buffer containing 25 mM Tris–HCl, 0.5% NP-40, 25 mM NaCl, and 10% glycerol and completed with the Protease Inhibitor Cocktail (1:100; P8340; Sigma-Aldrich), 10 mM sodium butyrate, 5 mM sodium fluoride, 1 mM sodium orthovanadate, and 100 mM phenylmethylsulfonyl fluoride. Protein quantifications were performed by Pierce BCA Protein Assay Kit (23227; Thermo Fisher Scientific). Protein G Sepharose 4 Fast Flow (17-0618-01; Merck) was incubated for 1 h with 4 $\mu g$ of antibody anti-human LC3 IgG (ab48394; Abcam), anti-human PACSIN2 IgG (TA325022; OriGene), or rabbit IgG, polyclonal, isotype control (ab37415; Abcam), blocked for 1 h with bovine serum albumin 1 mg/ml, and then incubated for 3 h with 500 $\mu g$ of cell lysates. After washing with lysis buffer, beads were eluted with sample buffer completed with 1 M dithiothreitol, boiled for 5 min at 95°C, and loaded on NuPAGE 10% Bis-Tris protein precast gels (NP0301BOX; Life Technologies) and 7% Tris-Tricine gels for immunoblotting of LC3-II and PACSIN2, respectively. In LS180 cells, for immunoblotting of LC3-II we used the NuPAGE 10% Bis-Tris protein precast gels (NP0301BOX; Life Technologies), whereas to detect PACSIN2 in the co-immunoprecipitation samples, we prepared 7% Tris-Tricine gels to improve separation between PACSIN2 and the heavy chains of the primary antibodies for the co-immunoprecipitations, which present similar molecular weights (PACSIN2 is detectable around 56 kD and antibody heavy chains around 55 kD).

## Drug sensitivity assay

NALM6 or LS180 cell lines were seeded as 100 $\mu l$ cell suspension in 96-well plates at 20,000 cells/well or 5,000 cells/well, respectively. Subsequently, 100 $\mu l$ per well of the substances tested (mercaptopurine or tunicamycin) at decreasing concentrations (Table S1) was added. Plates were incubated for 72 h in a humidified incubator (5% $CO_2$ at 37°C) in the presence of MTT (3-4,5-dimethylthiazol-2,5-diphenyl tetrazolium bromide, M2128; Sigma-Aldrich) 0.5 mg/ml for the last 4 h. After incubation, supernatants were removed and MTT precipitates were suspended in 100 $\mu l$ of dimethyl sulfoxide; absorbance at 570- to 630-nm wavelength was measured with a Microplate Reader EL311 (214891; BioTek Instruments). Results were expressed as percentages of survival according to the absorbance ratio between treated and untreated conditions (after blank subtraction), and EC50 values were then calculated on dose–response curves.

## Mitochondrial membrane potential measurement

LS180 cells were seeded in 96-well plates at 5,000 cells/well (100 $\mu l$). After 24 h, 100 $\mu l$ per well of mercaptopurine (final concentrations 1.25 and 2.5 $\mu M$) was added and incubated for 24 or 48 h; moreover, 5 $\mu M$ doxorubicin for 24 h was used as a positive control. 30 min before the end of incubation, wells were washed twice with 100 $\mu l$ PBS. Then, 50 nM of 3,3′-dihexyloxacarbocyanine iodide (DiOC6) probe in 100 $\mu l$ of PBS was added to samples and incubated for 30 min at 37°C and 5% $CO_2$. Finally, fluorescence was detected at 485/520-nm absorption/emission wavelength using the Microplate Reader EL311 (214891; BioTek Instruments).

## Thiopurine metabolite analysis

LS180 cells were seeded at $1.7 \times 10^6$ in 7 ml of RPMI medium and after 24 h were treated with mercaptopurine 2.5 $\mu M$ for 24 or 48 h. Then, cells were washed with PBS and centrifuged for 5 min at 600$g$; pellets were collected, suspended in 500 $\mu l$ of Milli-Q water, vortex-mixed, frozen, and thawed to obtain lysate samples. Thiopurine metabolites were quantified using a previously described method based on the high-performance liquid chromatography (HPLC-UV) analysis (Dervieux & Boulieu, 1998).

## Statistical analysis

Statistical analysis was performed with the software R (version 3.2.4) or PRISM (version 7.0). Associations were evaluated using a $t$ test, two-way ANOVA, and Pearson's correlation test. For all continuous variables, results are reported as means and standard error. All experiments were performed with at least three biological replicates.

# Supplementary Information

# Acknowledgements

$Atg7^{+/+}$, $Atg7^{-/-}$, $HeLa^{GFP-LC3}$, and $RAW^{GFP-LC3}$ cell lines were kindly provided by Dr. Green of the St. Jude Children's Research Hospital (SJCRH) in Memphis (USA). This work was supported by the Italian Ministry of Health, through the contribution given to the Institute for Maternal and Child Health IRCCS Burlo Garofolo, Trieste, Italy (RC 10/19).

## Author Contributions

G Zudeh: data curation, formal analysis, investigation, methodology, and writing—original draft.
R Franca: conceptualization, supervision, and writing—review and editing.
M Lucafò: conceptualization, supervision, and writing—review and editing.
EJ Bonten: investigation and writing—review and editing.
M Bramuzzo: resources and writing—review and editing.
R Sgarra: resources, methodology, and writing—review and editing.
C Lagatolla: resources, methodology, and writing—review and editing.
M Franzin: investigation and writing—review and editing.

WE Evans: conceptualization, resources, and writing—review and editing.

G Decorti: conceptualization, project administration, and writing—review and editing.

G Stocco: conceptualization, data curation, formal analysis, supervision, methodology, project administration, and writing—original draft, review, and editing.

## Conflict of Interest Statement

The authors declare that they have no conflict of interest.

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
