## [Reviewer comments · Life Science Alliance]

Life Science Alliance

PACSIN2 as a modulator of autophagy and mercaptopurine cytotoxicity in lymphoid and intestinal cells

Giulia Zudeh, Raffaella Franca, Marianna Lucafò, Erik Bonten, Matteo Bramuzzo, Riccardo Sgarra, Cristina Lagatolla, Martina Franzin, William Evans, Giuliana Decorti, and Gabriele Stocco

DOI: <https://doi.org/10.26508/lsa.202201610>

Corresponding author(s): Giuliana Decorti, Institute for Maternal and Child Health I.R.C.C.S. Burlo Garofolo

Review Timeline:

Submission Date:	2022-07-18
Editorial Decision:	2022-09-06
Revision Received:	2022-12-05
Editorial Decision:	2022-12-08
Revision Received:	2022-12-14
Accepted:	2022-12-15

Transaction Report:

September 6, 2022

Re: Life Science Alliance manuscript #LSA-2022-01610-T

Giuliana Decorti
Institute for Maternal and Child Health I.R.C.C.S. Burlo Garofolo
Department of Translational and Advanced Diagnostics
Via dell'Istria 65/1
Trieste 34137
ITALY

Dear Dr. Decorti,

Thank you for submitting your manuscript entitled "PACSLN2 as a modulator of autophagy and mercaptopurine cytotoxicity in lymphoid and intestinal cells" to Life Science Alliance. The manuscript was assessed by expert reviewers, whose comments are appended to this letter. We invite you to submit a revised manuscript addressing the Reviewer comments.

Thank you for this interesting contribution to Life Science Alliance. We are looking forward to receiving your revised manuscript.

Sincerely,

B. MANUSCRIPT ORGANIZATION AND FORMATTING:

Reviewer #1 (Comments to the Authors (Required)):

Prof. Decorti' s team identified that PACSIN2 inhibited autophagy and the novel role of PACSIN2 on IBD and thiopurine toxicity. This study is well designed, and many experiments strongly support the conclusion. Additionally, the manuscript is well written and data presentation is appropriate. This study deserves publication in the LSA.

Minor points

Down-regulation of PACSIN2 and ensuing enhancement of autophagy are observed in IBD mucosa. Are these molecular changes beneficial (adaptive) / harmful for the pathogenesis of IBD? Is PACSIN2 a promising target for treating/attenuating IBD? It will be better if the authors would discuss this point, especially clinical application of the current findings.

Reviewer #2 (Comments to the Authors (Required)):

In the manuscript entitled „PACSIN2 as a modulator of autophagy and mercaptopurine cytotoxicity: mechanisms in lymphoid and intestinal cells“, Giulia Zudeh and colleagues focus on the molecular and functional interplay between the protein PACSIN2, autophagy and thiopurine-mediated cytotoxicity. Although the authors are able to provide some very interesting and comprehensibly presented experimental data (for instance the direct protein-protein interaction between PACSIN2 and LC3), the current version of the manuscript has several relevant weaknesses and the clinical relevance of the drawn conclusions has hardly been validated. My concerns are listed below:

- All mechanistic/functional data have been acquired in vitro in cell lines, which are somehow limited in modelling the complex in vivo situation. The overall impact of the manuscript is therefore significantly limited by the fact that (with the exception of the potential inverse correlation between PACSIN2 and autophagy) none of the results are validated for their in vivo relevance.
- The fact that the authors can only speculate about the specific mechanism underlying the PACSIN2-mediated modulation of TPMT activity without providing experimental evidence represents another key limitation of the manuscript.
- In Figure 5a/b the authors describe the downregulation of PACSIN2 in inflamed versus non-inflamed colon tissue of IBD patients. In order to exclude that the observed regulation of PACSIN2 under inflammatory conditions is mainly due to an altered cellular composition of the inflamed colon tissue (e.g., increased immune cell infiltration), it would be important to confirm that intestinal epithelial cells and intestinal immune cells show comparable expression levels of PACSIN2 and/or to consider the cellular composition of the individual probes for the interpretation of results. Moreover, the statement in the text that there is a fold change of 2.2 in the PACSIN2 mRNA expression in the inflamed tissue is somehow misleading (as a fold change >1 implies increase instead of decrease) and does not seem to fit with the data depicted in Fig. 5a; the authors should comment on this. Finally, the described inverse correlation between PACSIN2 amount and autophagy levels has to be further confirmed by correlation analyses testing the relation between PACSIN2 and LC3-II levels in the individual biopsies.
- As mentioned in the introduction, PACSIN2 has already been identified as an inhibitor of autophagy in former studies. Thus, the novelty of the conclusion drawn from Figure 3 ("...taken together these results supported the hypothesis that PACSIN2 could play a role in the inhibition of autophagy.") is quite limited.
- In Figure 3, there is a significant difference in the LC3-I expression between MOCK and KD NALM6 cells, which cannot be observed in LS180 cells. How do the authors explain and interpret this different behavior of the two analyzed cell lines?
- It is hard to recognize the described decrease of SQSTM1/P62 levels in NALM6KD and LS180 KD cells in the representative Western Blot images depicted in Figure 3c/d. Moreover, it is stated in the text that NALM6 KD and LS180 KD cells showed an increased SQSTM1/P62 amount compared to MOCK (fold change NALM6: 1.57 {plus minus} 0.1, P= 0.047; LS180: 3.94 {plus minus} 1.8, P = 0.046) upon chloroquine treatment; however, these significant differences have not been indicated in the respective graphs.
- The numbers provided in the following sentence are not fully consistent and should be re-checked: "The cohort consists of 15 pediatric patients (mean age: 13.5 {plus minus} 0.75 years), equally distributed for gender (7 females (46.7 %) and 10 males (53.3 %), 11 presenting Crohn's disease (55%) and 9 ulcerative colitis (45%)."
- Regarding Figure 6, the authors should indicate in the figure legend how many times the depicted representative experiment presented was repeated. As the PACSIN2 Western blot in Fig. 6a shows several bands, it would be helpful to indicate which band can be assumed to be the specific PACSIN2 signal.

- Results depicted in Figure 7 need more careful interpretation and explanation directly in Results. Why do the two cell lines behave differently upon exposure to tunicamycin? First the authors mention that autophagy can be used as a survival mechanism to manage cellular stress; and afterwards they describe that NALM6 KD cells, which showed an increased level of autophagy, presented an increased sensitivity to the ER-stress inducer tunicamycin. How is this compatible? Via which mechanisms might PACSIN2 increase the tunicamycin-induced ER stress? As the authors discuss that the higher sensitivity of TPMT-overexpressing NALM6 cells is probably due to the supra-physiological accumulation of methylated metabolites, it would be better to change the term "mercaptopurine sensitivity" (which implicates therapeutic effects) to "mercaptopurine-induced cell toxicity". The hypothesis of a tissue specific role of PACSIN2 needs further concretization.
- Based on Figure 8, the authors describe the capacity of mercaptopurine to reduce autophagy in LS180 cells. However, in literature there exist good evidence for an autophagy-promoting effect of thiopurine drugs. While discussing their data in this context, the authors should also consider the aspect of physiological and clinical relevance.

Reviewer #1 (Comments to the Authors (Required)):

Prof. Decorti' s team identified that PACSIN2 inhibited autophagy and the novel role of PACSIN2 on IBD and thiopurine toxicity. This study is well designed, and many experiments strongly support the conclusion. Additionally, the manuscript is well written and data presentation is appropriate. This study deserves publication in the LSA.

Minor points

Down-regulation of PACSIN2 and ensuing enhancement of autophagy are observed in IBD mucosa. Are these molecular changes beneficial (adaptive) / harmful for the pathogenesis of IBD? Is PACSIN2 a promising target for treating/attenuating IBD? It will be better if the authors would discuss this point, especially clinical application of the current findings.

We thank the reviewer for this suggestion. Disrupted autophagy is considered a contributing factor for IBD pathogenesis and the drug modulation of this cellular mechanism is now under investigation as a new therapeutic strategy for IBD treatment. On these bases, it would be interesting to investigate the possible effect of PACSIN2 drug-mediated modulation on autophagy levels in intestinal IBD models. We have hypothesized that targeting PACSIN2 levels should result in beneficial molecular changes for IBD patients. We have added these considerations in the discussion part of the manuscript.

Reviewer #2 (Comments to the Authors (Required)):

In the manuscript entitled „PACSIN2 as a modulator of autophagy and mercaptopurine cytotoxicity: mechanisms in lymphoid and intestinal cells“, Giulia Zudeh and colleagues focus on the molecular and functional interplay between the protein PACSIN2, autophagy and thiopurine-mediated cytotoxicity. Although the authors are able to provide some very interesting and comprehensibly presented experimental data (for instance the direct protein-protein interaction between PACSIN2 and LC3), the current version of the manuscript has several relevant weaknesses and the clinical relevance of the drawn conclusions has hardly been validated.

My concerns are listed below:

- All mechanistic/functional data have been acquired *in vitro* in cell lines, which are somehow limited in modelling the complex *in vivo* situation. The overall impact of the manuscript is therefore significantly limited by the fact that (with the exception of the potential inverse correlation between PACSIN2 and autophagy) none of the results are validated for their *in vivo* relevance.

We thank the reviewer for this comment. We agree about the need of an *in vivo* validation of our results and we now acknowledge the lack of *in vivo* models as a major limitation of the current study in the revised discussion of the manuscript. On the bases of the encouraging results obtained in this study, we plan to perform *in vivo* experiments in the future, to explore the causality between PACSIN2 modulation and autophagy also *in vivo*. We tried to improve our manuscript with additional *ex vivo* data on intestinal biopsies, further highlighting a correlation between PACSIN2 and LC3-II (Figure 5a and supplementary Figure 4 in the revised manuscript).

- The fact that the authors can only speculate about the specific mechanism underlying the PACSIN2-mediated modulation of TPMT activity without providing experimental evidence represents another key limitation of the manuscript.

We thank the reviewer for this comment. Because the molecular mechanism at the basis of PACSIN2 modulation of TPMT is still uncharacterized, we performed some experiments in NALM6 cell overexpressing TPMT to evaluate if autophagy could be involved in this process; however, the obtained results indicated that autophagy is not involved in the PACSIN2 modulation of TPMT. We have underlined in the discussion of the manuscript the need of a further investigation to evaluate the molecular mechanism at the basis of the PACSIN2 modulation on TPMT.

- In Figure 5a/b the authors describe the downregulation of PACSIN2 in inflamed versus non-inflamed colon tissue of IBD patients. In order to exclude that the observed regulation of PACSIN2 under inflammatory conditions is mainly due to an altered cellular composition of the inflamed colon tissue (e.g., increased immune cell infiltration), it would be important to confirm that intestinal epithelial cells and intestinal immune cells show comparable expression levels of PACSIN2 and/or to consider the cellular composition of the individual probes for the interpretation of results. -Moreover, the statement in the text that there is a fold change of 2.2 in the PACSIN2 mRNA expression in the inflamed tissue is somehow misleading (as a fold change >1 implies increase instead of decrease) and does not seem to fit with the data depicted in Fig. 5a; the authors should comment on this.

-Finally, the described inverse correlation between PACSIN2 amount and autophagy levels has to be further confirmed by correlation analyses testing the relation between PACSIN2 and LC3-II levels in the individual biopsies.

We thank the reviewer for these comments and revised the manuscript accordingly.

In order to verify whether intestinal epithelial cells and intestinal immune cells show comparable expression levels of PACSIN2, we evaluated PACSIN2 mRNA amount in six intestinal organoids samples and in six IBD patient's whole blood samples. No statistically significant difference in the PACSIN2 level was observed (supplementary figure 5a of the revised manuscript). Furthermore, we reported in the supplementary figure 5b different comparisons of PACSIN2 transcriptional level found in intestinal biopsies cells, such as colon endothelium and fibroblast cells and many immune system's cell types, which may be infiltrated in the intestinal biopsies using the "IBD Transcriptome and Metatranscriptome Meta-Analysis" (IBD TaMMA) database (doi.org/10.1038/s43588-021-00114-y); consistently, no statistically significant difference was detected.

As the reviewer suggested, we checked the fold change numbers of the Figure 5a and corrected them. We apologize for the mistake. Moreover, we modified the image present in the figure 5b because we added two samples in these analyses to reply to the reviewer comment about the correlation between PACSIN2 and LC3; we also corrected the respective data values in the result part of the manuscript and in the figure legend accordingly.

Finally, as requested by the reviewer, an *ex vivo* correlation analysis between PACSIN2 and LC3-II levels in the intestinal biopsies was performed, showing a significant correlation between them, as now shown in supplementary Figure 4.

We added these results in the manuscript text and included the corresponding figures in the supplementary section.

- A mentioned in the introduction, PACSIN2 has already be identified as an inhibitor of autophagy in former studies. Thus, the novelty of the conclusion drawn from Figure 3 ("...taken together these results supported the hypothesis that PACSIN2 could play a role in the inhibition of autophagy.") is quite limited.

Thank the reviewer for the suggestion. So far, involvement of PACSIN2 in the autophagy machinery has only been demonstrated with agnostic approaches and without clarifying the direction of the effect, whereas experimental confirmations have been lacking. Moreover, two previous studies detected the involvement of PACSIN1 in the regulation of autophagy machinery; in particular, Szyniarowski and collaborators demonstrated that PACSIN1 depletion affected autophagy, whereas Oe and colleagues found the modulation of the last phases of the autophagic flux by PACSIN1, which played a role during the autophagosomes fusion with lysosomes. We highlighted the novelty of our study rephrasing the suggested sentence. In particular, the current study confirmed the involvement of PACSIN2 in the autophagy modulation and also demonstrated the direction of PACSIN2 effect on the autophagy regulation.

We explained these considerations in the introduction and the discussion of the manuscript.

- In Figure 3, there is a significant difference in the LC3-I expression between MOCK and KD NALM6 cells, which cannot be observed in LS180 cells. How do the authors explain and interpret this different behavior of the two analyzed cell lines?

We thank the reviewer for the question. The different amount of this isoform of LC3 between NALM6 and LS180 cells could be due to the different origin of these cell lines, as showed in the human protein ATLAS (Human Protein Atlas proteintlas.org), which indicates a lower expression of LC3 in the lymphoid tissue compared to intestine and could be explained by an alternative epigenetic regulation of LC3 in different tissues; consistently, our western blot results for LC3 indicated a higher autophagy basal level in LS180 cells compared to NALM6.

The presence of *PACSN2* KD stimulated the autophagy induction and the consequent LC3-II conversion from LC3-I in both NALM6 and LS180 cells. The increase LC3-I conversion to LC3-II observed only in NALM6 KD cells, could be due to the lower amount of LC3 in this tissue, an consequently the autophagosome formation could stimulate LC3 transcription, as observed in other studies with other autophagy modulators (doi.org/10.1038/s41419-017-0073-9), leading to an increase of LC3-I in this cells. We added a comment on this in the discussion part of the manuscript.

- It is hard to recognize the described decrease of SQSTM1/P62 levels in NALM 6KD and LS180 KD cells in the representative Western Blot images depicted in Figure 3c/d. Moreover, it is stated in the text that NALM6 KD and LS180 KD cells showed an increased SQSTM1/P62 amount compared to MOCK (fold change NALM6: 1.57 {plus minus} 0.1, P= 0.047; LS180: 3.94 {plus minus} 1.8, P = 0.046) upon chloroquine treatment; however, these significant differences have not been indicated in the respective graphs.

We thank the reviewer for this comment and apologize for the incomplete graph. Accordingly, we changed the figure 3 and modified the corresponding sentence in the text, to explain our results in a clearer way.

- The numbers provided in the following sentence are not fully consistent and should be re-checked: "The cohort consists of 15 pediatric patients (mean age: 13.5 {plus minus} 0.75 years), equally distributed for gender (7 females (46.7 %) and 10 males (53.3 %), 11 presenting Crohn's disease (55%) and 9 ulcerative colitis (45%)."

We thank the reviewer for the suggestion. We modified this part of the text, adding the samples used for the additional analyses performed during the manuscript revision and adjusted the data present in the paragraph about patient's cohort; we moved this information in the material and methods section in the "patient's sample" paragraph.

- Regarding Figure 6, the authors should indicate in the figure legend how many times the depicted representative experiment presented was repeated. As the PACSN2 Western blot in Fig. 6a shows several bands, it would be helpful to indicate which band can be assumed to be the specific PACSN2 signal.

We thank the reviewer for this comment and on the basis of the suggestion, we added the number of replicates in the figure legend and indicated better the specific PACSN2 band in Figure 6a.

- Results depicted in Figure 7 need more careful interpretation and explanation directly in Results. Why do the two cell lines behave differently upon exposure to tunicamycin? First the authors mention that autophagy can be used as a survival mechanism to manage cellular stress; and afterwards they describe that NALM6 KD cells, which showed an increased level of autophagy,

presented an increased sensitivity to the ER-stress inducer tunicamycin. How is this compatible? Via which mechanisms might PACSIN2 increase the tunicamycin-induced ER stress? As the authors discuss that the higher sensitivity of TPMT-overexpressing NALM6 cells is probably due to the supra-physiological accumulation of methylated metabolites, it would be better to change the term "mercaptapurine sensitivity" (which implicates therapeutic effects) to "mercaptapurine-induced cell toxicity". The hypothesis of a tissue specific role of PACSIN2 needs further concretization.

We thank the reviewer for this comment.

Despite of autophagy being a survival mechanism, in presence of elevated stress conditions, cells may move from autophagy to apoptosis through mechanisms involving the endoplasmic reticulum (ER). Tunicamycin is an ER stress inducer, stimulating the accumulation of unfolded proteins, which need to be removed by autophagy. Because cells with *PACSIN2* KD presented higher autophagy levels (higher LC3 and lower SQSTM1/P62 levels) due to the silencing of this gene, the exposure to tunicamycin represented one further stress source and may impact on cell vitality. Indeed, the MTT assay results showed that LS180 cells were more sensible to tunicamycin than NALM6 cells; it has to be noted that LS180 cells showed a higher basal autophagy level compared to NALM6 and the presence of a further stress source, such as the tunicamycin exposure, might affect cell vitality stimulating a further cell death both in LS180 MOCK and KD cells, which did not permit to detect the possible effect of *PACSIN2* KD in this intestinal model. We modified the discussion adding a comment about this consideration and also added a sentence in the result paragraph.

As the reviewer suggested, we changed the term "drug sensitivity" to "drug cytotoxicity" for both mercaptopurine and tunicamycin cytotoxicity results.

In the current study we used the *PACSIN2* KD as an *in vitro* model to evaluate the mechanisms which undergoes to the effect of *PACSIN2* in the landscape of thiopurines cytotoxicity. The obtained cytotoxicity results were in accordance to the patient's results, because the *PACSIN2* KD increased mercaptopurine-induced cytotoxicity in the intestinal cells and not in the lymphoid cells. Therefore, these *in vitro* results supported the hypothesis of a possible tissue specific role of *PACSIN2* in the intestine; however, further investigations are needed to validate this result and to clarify the molecular mechanism at the basis of these evidences. We better explained these considerations in the discussion part of the manuscript.

- Based on Figure 8, the authors describe the capacity of mercaptopurine to reduce autophagy in LS180 cells. However, in literature there exist good evidence for an autophagy-promoting effect of thiopurine drugs. While discussing their data in this context, the authors should also consider the aspect of physiological and clinical relevance.

We thank the reviewer for the comment and revised the discussion accordingly, adding some considerations on the clinical relevance of our results. We highlighted particularly that disrupted autophagy is considered a contributing factor for IBD pathogenesis and the drug modulation of this cellular mechanism is now under investigation as a new therapeutic strategy for IBD treatment. On these bases, it would be interesting to investigate the possible effect of *PACSIN2* drug-mediated modulation on autophagy levels in intestinal IBD models. We have hypothesized that targeting *PACSIN2* levels should result in beneficial molecular changes for IBD patients. We commented the discordance between our results and those obtained by Chaabane and colleagues, which may be due to the different experimental conditions used in the two studies, such as the tested mercaptopurine's concentrations.

December 8, 2022

RE: Life Science Alliance Manuscript #LSA-2022-01610-TR

Prof. Giuliana Decorti
Institute for Maternal and Child Health I.R.C.C.S. Burlo Garofolo
Department of Translational and Advanced Diagnostics
Via dell'Istria 65/1
Trieste 34137
Italy

Dear Dr. Decorti,

Thank you for submitting your revised manuscript entitled "PACSIN2 as a modulator of autophagy and mercaptopurine cytotoxicity in lymphoid and intestinal cells". We would be happy to publish your paper in Life Science Alliance pending final revisions necessary to meet our formatting guidelines.

- please add ORCID ID for corresponding author-you should have received instructions on how to do so
- please add the Twitter handle of your host institute/organization as well as your own or/and one of the authors in our system
- please make sure that all the authors' names in the manuscript match the names in our system

Figure Check:

- In the blots in Figure 3A and Figure, please indicate the splice between lanes by a black line rather than the white blank space. Also mention in the figure legends that the line indicates a splice. We'd also recommend uploading the raw blots as Source Data.
- There also appears to be a splice in the actin blot of Figure 10B. If so, please indicate this in the figure and legend, and provide the raw blot as Source Data.

A. FINAL FILES:

B. MANUSCRIPT ORGANIZATION AND FORMATTING:

Sincerely,

December 15, 2022

RE: Life Science Alliance Manuscript #LSA-2022-01610-TRR

Prof. Giuliana Decorti
Institute for Maternal and Child Health I.R.C.C.S. Burlo Garofolo
Department of Translational and Advanced Diagnostics
Via dell'Istria 65/1
Trieste 34137
Italy

Dear Dr. Decorti,

Thank you for submitting your Research Article entitled "PACSIN2 as a modulator of autophagy and mercaptopurine cytotoxicity in lymphoid and intestinal cells". It is a pleasure to let you know that your manuscript is now accepted for publication in Life Science Alliance. Congratulations on this interesting work.

DISTRIBUTION OF MATERIALS:

Again, congratulations on a very nice paper. I hope you found the review process to be constructive and are pleased with how the manuscript was handled editorially. We look forward to future exciting submissions from your lab.

Sincerely,
